# Finite-Time Bounds for Average-Reward Fitted Q-Iteration

**Jongmin Lee**
Seoul National University
Department of Mathematical Sciences
dlwhd2000@snu.ac.kr

**Ernest K. Ryu**
UCLA
Department of Mathematics
eryu@math.ucla.edu

## Abstract

Although there is an extensive body of work characterizing the sample complexity of discounted-return offline RL with function approximations, prior work on the average-reward setting has received significantly less attention, and existing approaches rely on restrictive assumptions, such as ergodicity or linearity of the MDP. In this work, we establish the first sample complexity results for average-reward offline RL with function approximation for weakly communicating MDPs, a much milder assumption. To this end, we introduce Anchored Fitted Q-Iteration, which combines the standard Fitted Q-Iteration with an anchor mechanism. We show that the anchor, which can be interpreted as a form of weight decay, is crucial for enabling finite-time analysis in the average-reward setting. We also extend our finite-time analysis to the setup where the dataset is generated from a single-trajectory rather than IID transitions, again leveraging the anchor mechanism.

## 1 Introduction

The goal of offline Reinforcement Learning (RL) is to find a near-optimal policy using a precollected dataset without any direct interaction with the environment. Characterizing the sample complexity for finding an $\epsilon$-optimal policy using function approximation under assumptions that the offline data has sufficient coverage over the whole state-action space has been an active area of theoretical RL research. However, more prior work focuses on the discounted cumulative reward setup, and research on obtaining sample complexity in the average reward has been limited due to the absence of the discount factor and the complexity of the Bellman equation. Specifically, all prior works with function approximation rely on restrictive assumptions such as ergodicity or linearity of the MDP.

Although theoretical RL research often focuses on the discounted return setup due to the theoretical convenience offered by the discount factor and the simpler Bellman equation, many practical scenarios are more naturally modeled as agents aim to maximize the average reward. In fact, many practical RL applications do not use discounting at all. These considerations make the sample complexity of average-reward RL relevant, despite the additional technical challenges this setting presents.

**Contribution.** In this work, we introduce the Anchored Fitted Q-Iteration and establish the sample complexity on average reward MDPs with general function approximation for weakly communicating MDPs for the first time. We consider the cases with IID data and with single-trajectory data. Then, we show that by using the relative normalization mechanism from the classical relative value iteration, we can further improve the sample complexity.

39th Conference on Neural Information Processing Systems (NeurIPS 2025).

| Prior works | MDP class | dataset | Coverage coefficient |
|---|---|---|---|
| Ozdaglar et al. [63] | ergodic* | IID samples | partial |
| Gabbianelli et al. [30] | unichain* (+ linear) | IID samples | partial |
| Our work | weakly communicating* | IID samples | full |
| Our work | weakly communicating* | $\beta$-mixing single-trajectory | full |

Table 1: Comparison of analyses of offline average-reward MDPs. Our work, which assumes the MDP is weakly communicating, significantly relaxes the structural assumption on the MDP compared to prior work. (Clarification*: Ergodic, unichain, and weakly communicating are respectively the standard MDP classes for which the results of [63], [30], and our work apply. However, the precise conditions are slightly more general in each case. See Section 1.2 for detailed definitions.)

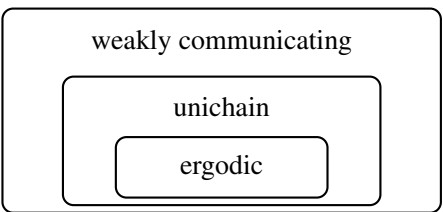

Figure 1: The MDP classes satisfy the inclusion: ergodic $\subset$ unichain $\subset$ weakly communicating

## 1.1 Preliminaries and notations

We briefly review the basic notions of average-reward Markov decision processes (MDPs) and reinforcement learning (RL) and refer the readers to standard references for further details [66, 7, 78].

**Average-reward MDP.** Let $\mathcal{M}(\mathcal{X})$ be the space of probability distributions over $\mathcal{X}$ and $\mathcal{F}(\mathcal{X})$ be the space of bounded real-valued functions over $\mathcal{X}$. Write $(\mathcal{S}, \mathcal{A}, P, r)$ to denote an infinite-horizon undiscounted MDP with finite state space $\mathcal{S}$, finite action space $\mathcal{A}$, transition matrix $P: \mathcal{S} \times \mathcal{A} \to \mathcal{M}(\mathcal{S})$, bounded reward $r: \mathcal{S} \times \mathcal{A} \to [-R, R]$. Denote $\pi: \mathcal{S} \to \mathcal{M}(\mathcal{A})$ for a policy, $g^\pi(s,a) = \liminf_{T \to \infty} \frac{1}{T} \mathbb{E}_\pi \left[ \sum_{t=1}^T r(s_t, a_t) \mid s_0 = s, a_0 = a \right]$ for the average-reward of a policy $\pi$ given an initial state-action pair $(s, a)$, where $\mathbb{E}_\pi$ denotes the expectation over all trajectories $(s_0, a_0, s_1, a_1, \ldots, s_T, a_T)$ induced by $P$ and $\pi$.

We say $\pi_\star$ is an optimal policy if $g^{\pi_\star}(s,a) = \max_\pi g^\pi(s,a)$ for all $s \in \mathcal{S}$ and $a \in \mathcal{A}$, and we say $g^{\pi_\star}$ is the optimal average reward. (The optimal policy and optimal average reward exists for finite state-action space [66, Theorem 9.1.8].) We say $\pi$ is an $\epsilon$-optimal policy if $\|g^{\pi_\star} - g^\pi\|_\infty \leq \epsilon$. Define $\mathcal{P}^\pi$ as

$$\mathcal{P}^\pi((s,a) \to (s',a')) = \mathrm{Prob}((s,a) \to (s',a') \mid s' \sim P(\cdot \mid s,a), a' \sim \pi(\cdot \mid s')),$$

the transition matrix induced by policy $\pi$. Then, $(\mathcal{P}^\pi Q)(s,a) = \mathbb{E}_{a' \sim \pi(\cdot \mid s'), s' \sim P(\cdot \mid s,a)}[Q(s',a')]$ for $Q \in \mathcal{F}(\mathcal{S} \times \mathcal{A})$. Define the weighted $L_p$-norm of $Q \in \mathcal{F}(\mathcal{S} \times \mathcal{A})$ under state-action distribution $\rho$ as $\|Q\|_{p,\rho} = [\mathbb{E}_{(s,a) \sim \rho}|Q(s,a)|^p]^{1/p}$ for $p \geq 1$.

**Coverage coefficient.** A coverage coefficient quantifies the shift between the distribution of the offline data and the distribution induced by policies [61, 18]. Loosely speaking, the *full coverage* assumption, as stated in Table 1, assumes that the offline data sufficiently explores the whole state-action space regardless of policy [4, 92], while *partial coverage* only requires the offline data to sufficiently explore the state-action pairs that an optimal policy would visit [94, 42]. These types of assumptions are fundamentally necessary for the complexity analysis of offline RL [18], and different works use different types of coverage coefficients (cf. [80, 71]). The coverage coefficient we use is defined in Section 3.

**Value Iteration.** Given an undiscounted MDP $(\mathcal{S}, \mathcal{A}, P, r)$, the Bellman optimality operator $T$ is

$$TQ(s,a) = r(s,a) + \mathbb{E}_{s' \sim P(\cdot \mid s,a)} \left[ \max_{a' \in \mathcal{A}} Q(s', a') \right]$$

for all $s \in \mathcal{S}$ and $a \in \mathcal{A}$. We define the standard Value Iteration (VI) as

$$Q^k = TQ^{k-1} \qquad \text{for } k = 1, 2, \ldots, K,$$

where $Q^0$ is an initial point.

**MDP classes.** MDPs are classified according to the structure of the transition matrices. (For definitions on irreducible classes, recurrent classes, transient states, and aperiodicity of transition matrices, refer to [66, Appendix A.2].) An MDP is *ergodic* if the transition matrices induced by every policy $\pi$ has a single recurrent class and is aperiodic. An MDP is *unichain* if the transition matrices induced by every policy $\pi$ has a single recurrent class plus a possibly empty set of transient states. An MDP is *weakly communicating* if there is a set of states where every state in the set is accessible from every other state in that set under some policy, plus a possibly empty set of states that are transient for all policies. Otherwise, in general, an MDP is *multichain*. We note that classification of MDPs is crucial in the analyses of average-reward MDPs [96, 97, 88, 50].

## 1.2 Conditions of Prior works

In Table 1, we remarked that the precise conditions on the MDPs are slightly general than ergodic, unichain, and weakly communicating. In this section, we state the precise conditions.

**Uniform mixing [63].** The *uniform mixing* condition assumes that there exist positive $t_{mix} \in \mathbb{N}$ (which does not depend on $\pi$ and $\rho$) such that $\|\rho^\top (\mathcal{P}^\pi)^t - \nu^\pi\|_1 \leq 1/2$ for all $t \geq t_{max}$ for any policy $\pi$ and initial distribution $\rho$, where $\nu^\pi$ is the stationary distribution of $\mathcal{P}^\pi$. (This condition requires that the stationary distribution $\nu^\pi$ is unique for all $\pi$.) The prior work [63] uses this assumption for its analysis of average-reward MDPs. Ergodic MDPs satisfy the uniform mixing condition [13, 52], but unichain MDPs do not [66, Example 8.2.1].

**All-policy Bellman equation and linear MDPs [30].** The *all-policy Bellman equation* states that for any policy $\pi$, the average reward $g^\pi$ does not depend on $(s,a)$ and there exist a $Q^\pi : \mathcal{S} \times \mathcal{A} \to \mathbb{R}$ such that

$$r(s,a) + \mathbb{E}_{a' \sim \pi(\cdot \mid s'), s' \sim P(\cdot \mid s,a)} \left[ Q^\pi(s', a') \right] = Q^\pi(s,a) + g^\pi$$

for all $s \in \mathcal{S}$ and $a \in \mathcal{A}$. The prior work [30] uses this assumption for its analysis of average-reward MDP. Unichain MDPs satisfy the all-policy Bellman equation while weakly communicating MDPs, in general, do not [66, Section 8.4]. Also, the uniform mixing condition implies the all-policy Bellman equation [88, Lemma 6].

An MDP is *linear* if there exist $\phi : \mathcal{S} \times \mathcal{A} \to \mathbb{R}^d$, $\psi : \mathcal{S} \to \mathbb{R}^d$, and $w \in \mathbb{R}^d$ such that

$$r(s,a) = \langle \phi(s,a), w \rangle, \quad P(s' \mid s,a) = \langle \phi(s,a), \psi(s') \rangle.$$

The linear MDP assumption is often used for theoretical analyses [39, 30], but it requires knowledge of the mapping $\phi$ and $\psi$ and often fails to hold in practice [32, 82].

**Bellman optimality equation (our work).**

**Assumption 1** (Bellman optimality equation). *The optimal average reward $g^{\pi_\star}$ does not depend on $(s,a)$ and there exist a $Q^{\pi_\star} : \mathcal{S} \times \mathcal{A} \to \mathbb{R}$ such that*

$$r(s,a) + \mathbb{E}_{s' \sim P(\cdot \mid s,a)} \left[ \max_{a'} Q^{\pi_\star}(s', a') \right] = Q^{\pi_\star}(s,a) + g^{\pi_\star},$$

*for all $s \in \mathcal{S}$ and $a \in \mathcal{A}$.*

A policy $\pi_\star$ satisfying the Bellman optimality equation is an optimal policy [88, Section 2]. The all-policy Bellman equation implies the Bellman optimality equation, and the weakly communicating condition of MDPs also implies the Bellman optimality equation [66, Theorem 8.3.2, 8.4.1].

## 2  Anchored Fitted Q-Iteration

Consider the offline RL setup with precollected dataset $D = \{s_i, a_i, r_i, s_i'\}_{i=1}^n$, where $r_i = r(s_i, a_i)$ and $s_i' \sim P(\cdot \,|\, s_i, a_i)$. Let $\mathcal{F}$ be a nonempty function space to approximate $Q$-value. We now introduce our novel algorithm, *Anchored Fitted Q-Iteration (Anc-F-QI)*.

---

**Algorithm 1** Anchored Fitted Q-Iteration $(D, K, \{\mathcal{F}_i\}_{i=1}^K \{\lambda_i\}_{i=1}^K)$

---

**Input**: $D = \{s_i, a_i, r_i, s_i'\}_{i=1}^n$, $f_0 = 0$, $K \geq 1$, $\{\lambda_i\}_{i=1}^K \subset (0, 1)$
**for** $k = 0, 1, \ldots, K-1$ **do**
$\quad \hat{T}f_k = \mathrm{argmin}_{f \in \mathcal{F}_{k+1}} \sum_{i=1}^n \left( f(s_i, a_i) - r_i - \max_{a \in \mathcal{A}} f_k(s_i', a) \right)^2$
$\quad f_{k+1} = (1 - \lambda_{k+1})f_0 + \lambda_{k+1}\hat{T}f_k \qquad \triangleright$ With $f_0 = 0$, this is weight decay
**end for**
$\pi(a \,|\, s) = \mathrm{argmax}_{a \in \mathcal{A}} f_K(s, a)$
**Output** $\pi, f_K$

---

In Section 4, we present our sample complexity results for Anc-F-QI. Roughly speaking, Theorem 1 establishes $\tilde{\mathcal{O}}(1/\epsilon^6)$ sample complexity with IID data and Theorem 2 establishes $\tilde{\mathcal{O}}(1/\epsilon^{12})$ sample complexity with $\beta$-mixing single-trajectory data. In Section 5, to further improve the sample complexity with the *Relative Anchored* Fitted-Q Iteration, establishing $\tilde{\mathcal{O}}(1/\epsilon^4)$ and $\tilde{\mathcal{O}}(1/\epsilon^8)$ sample complexities for IID and single-trajectory data, respectively.

### 2.1  The anchor mechanism and weight decay

Our method Anc-F-QI stated as Algorithm 1 consists of two main components. The first component is the first line of the for-loop, the classical Fitted Q-Iteration [25, 61] step without discount factor. Its goal is to find the function $\hat{T}f_k \approx Tf_k$, where $T$ is the Bellman operator. However, unlike Fitted Q-Iteration in the discounted cumulative reward case, $\hat{T}f_k \approx Tf_k$ is not enough to establish a finite sample complexity in the average-reward setup. In the tabular case where the Fitted Q-Iteration reduces to Value Iteration (VI), it is known that VI might not converge. Specifically, there exists an average-reward MDP such that the policy error of VI does not converge to zero [22, Example 4]. Even if an aperiodicity condition is assumed, VI guarantees only asymptotic convergence without any known explicit convergence rate in the average-reward setup [66, Theorem 9.4.5].

Recently, Anchored Value Iteration (Anc-VI) was proposed to obtain finite-time bounds of policy error for average-reward MDPs [14, 50, 48]. Particularly, the *Anchored Q-Value Iteration* is

$$Q^k = (1 - \lambda_k)Q^0 + \lambda_k TQ^{k-1} \qquad \text{for } k = 1, 2, \ldots. \qquad \text{(Anc-QI)}$$

where $\lambda_k$ parameter is to be chosen. Compared to the standard VI, Anc-QI obtains the next iterate as a convex combination between the output of $T$ and the *starting point* $Q^0$. We call the $(1 - \lambda_k)Q_0$ term the *anchor term* since it serves to pull the iterates back toward the starting point $Q_0$. With this Anc-VI, [50] establish non-asymptotic convergence in the average-reward setup. Specifically, Anc-VI exhibits the $O(1/k)$-rate in terms of policy error [50][Theorem 2 and Corollary 2] without any restrictions on the MDP.

This anchoring mechanism, classically also known as the Halpern iteration [33], has been widely studied in minimax optimization and fixed-point problems [70, 55, 65, 20, 93]. In the context of reinforcement learning, [49, 50] applied the anchoring mechanism to VIs for cumulative-return and average-reward MDPs under the tabular setting, and [14, 48] applied the anchoring mechanism to Q-Value Iteration for cumulative-return and average reward MDPs under the generative model setting.

In this work, we combine Fitted Q-Iteration with anchoring, as shown in the second line of the for-loop of Algorithm 1, and establish finite-time bounds on the sample complexity.

### 2.2  Assumptions on the function space $\mathcal{F}$

**Assumption 2** (existence of argmin). *In Anc-F-QI, the argmin defining $\hat{T}f_k$ exist for $k = 0, \ldots, K-1$.*

This assumption is needed for the regression step of Algorithm 1 to be well defined.

**Assumption 3** (star-shaped function space). *If $f \in \mathcal{F}, \eta f \in \mathcal{F}$ for all $\eta \in [0, 1]$.*

This assumption implies that the anchor step of Anc-F-QI to be well defined. Star-shaped function space is a classical notion that relaxes convexity [31, 34, 51], and if $\mathcal{F}$ corresponds to a parametrized neural network with a linear layer as the output layer, $\mathcal{F}$ is star-shaped.

**Definition 1** (Inherent Bellman error). *Define $\epsilon_B(\mathcal{F}, \mathcal{F}') = \max_{f \in \mathcal{F}} \min_{f' \in \mathcal{F}'} \|f' - Tf\|$ as the inherent Bellman error with respect to the norm $\|\cdot\|$.*

The inherent Bellman error $\epsilon_B$ quantifies the error due to the function spaces $\mathcal{F}, \mathcal{F}'$ in approximating the output of the Bellman operator [61, 3, 18]. Note that if the function spaces $\mathcal{F}, \mathcal{F}'$ are bounded (in the $\|\cdot\|_\infty$-norm), then $\epsilon_B$ is also bounded.

**Assumption 4** (Bellman completeness). *$\epsilon_B(\mathcal{F}, \mathcal{F}') = 0$, where $\epsilon_B$ the is inherent Bellman error.*

Bellman completeness states that if $f \in \mathcal{F}$, then $Tf \in \mathcal{F}'$. I.e., $\mathcal{F}, \mathcal{F}'$ are closed under the Bellman operator. Although the Bellman completeness assumption is seemingly strong, it is often considered in sample complexity analyses in the offline RL literature [18, 26]. In fact, the Bellman completeness condition is fundamental in the sense that the prior work [28] showed that a polynomial sample complexity cannot be established without Bellman completeness assumption.

# 3 Approximate Anchored Q-Value Iteration

In this section, we conduct an $L_p$ bound analysis that will later be used to establish the main sample complexity results of Sections 4 and 5. Define the *Approximate Anchored Q-Value Iteration* as

$$Q^k = (1 - \lambda_k)Q^0 + \lambda_k(TQ^{k-1} + \epsilon_k) \qquad \text{(Apx-Anc-QI)}$$

for $k = 1, 2, \ldots, K$, where $T$ is the Bellman operator, $Q^0 \in \mathbb{R}^n$ is a starting point, and $\epsilon_k$ represents the evaluation error of $TQ^{k-1}$. We choose $\lambda_k = \frac{k}{k+2}$ for $k = 1, \ldots, K$, motivated by [70, 20].

We now establish a convergence analysis of Apx-Anc-QI based on $L_p$ bounds of $\epsilon_k$. Similar to the prior work [59, 60, 26], we assume the following coverage coefficient for our analysis.

**Assumption 5** (uniform stochastic transition). *For a given distribution $\mu$ on $\mathcal{S} \times \mathcal{A}$,*

$$C_\mu \overset{\text{def}}{=} \sup_{s,a,\pi} \left\| \frac{\mathcal{P}^\pi(\cdot \mid s, a)}{\mu(\cdot)} \right\|_\infty < \infty.$$

**Assumption 6** (uniform future state distribution). *For given distributions $\mu$ and $\rho$ on $\mathcal{S} \times \mathcal{A}$,*

$$C_{\mu,\rho} \overset{\text{def}}{=} \sup_{\pi_1, \pi_2, \ldots \pi_k} \left\| \frac{\rho^\top \mathcal{P}^{\pi_1} \mathcal{P}^{\pi_2} \ldots \mathcal{P}^{\pi_k}(\cdot)}{\mu(\cdot)} \right\|_\infty < \infty,$$

*where $\pi_1, \pi_2, \ldots \pi_k$ represents an arbitrary sequence of policies.*

The coverage coefficients measure the mismatch between the distribution of offline data and the distribution induced by the transition matrices and initial distributions. We note that Assumption 5 implies Assumption 6 with $C_{\mu,\rho} \leq C_\mu$ [61, Section 5].

**Proposition 1.** *Let $p \in [1, \infty]$, and let $\mu$ and $\rho$ be distributions on $\mathcal{S} \times \mathcal{A}$. Under Assumption 1 and 5 (Bellman optimality equation, uniform stochastic transition), the policy error of Apx-Anc-QI with $\lambda_k = \frac{k}{k+2}$ satisfies*

$$\|g^{\pi_\star} - g^{\pi_K}\|_\infty \leq C_\mu^{1/p} \frac{8}{K+2} \|Q^{\pi_\star} - Q^0\|_{p,\mu} + C_\mu^{1/p} \frac{2K}{3} \max_{1 \leq k \leq K} \|\epsilon_k\|_{p,\mu}.$$

*Similarly, under Assumption 1 and 6 (Bellman optimality equation, uniform future state distribution), the policy error of Apx-Anc-QI with $\lambda_k = \frac{k}{k+2}$ satisfies*

$$\|g^{\pi_\star} - g^{\pi_K}\|_{p,\rho} \leq C_{\mu,\rho}^{1/p} \frac{8}{K+2} \|Q^{\pi_\star} - Q^0\|_{p,\mu} + C_{\mu,\rho}^{1/p} \frac{2K}{3} \max_{1 \leq k \leq K} \|\epsilon_k\|_{p,\mu}.$$

To clarify, the $g$-, $Q$-, and $\epsilon$-terms in Proposition 1 are functions of $(s, a)$ and the norms $\|\cdot\|_{p,\rho}$ and $\|\cdot\|_{p,\mu}$ are taking expectations with respect to the distributions $\rho$ and $\mu$.

The bounds of Proposition 1 serve as the technical crux of our sample complexity results later presented in Theorems 1, 2, 3, and 4. In the bound, the first term decreases with order $\mathcal{O}(1/K)$ but the second error term increases with order $\Theta(K)$. Therefore, our subsequent arguments will ensure $\|\epsilon_k\|_{p,\mu} = \mathcal{O}(1/K^2)$ by using sufficient offline samples.

# 4 Sample complexity of Anchored Fitted Q-Iteration

We now present sample complexity analyses of Anc-F-QI with IID and single-trajectory data.

## 4.1 Range of function space $\mathcal{F}$

Before analyzing the complexity of those, we explain our issue on range of function space in average-reward setup and our choice of function space.

When considering Fitted Q-Iteration in the discounted reward setup, the functions are often assumed to be bounded by $\|Q_\gamma^\star\|_\infty$ [61, 18], where $Q_\gamma^\star$ is optimal state-action function with discount factor $\gamma$, since $\|f\|_\infty \leq \|Q_\gamma^\star\|_\infty$ implies $\|Tf\|_\infty \leq \|Q_\gamma^\star\|_\infty$. In the average-reward setup (without discounting), this property does not hold, and the Fitted Q-Iteration is expected to produce an unbounded sequence of functions. To address this issue, we allow the range of the function space to increase with each iteration.

**Assumption 7** (increasing function range). *Let $\mathcal{F}_0 = \{0\}$ and $\mathcal{F}_k \subset \{f : \mathcal{S} \times \mathcal{A} \to [-kR, kR] : f \in B(\mathcal{S} \times A)\}$ and $f_k \in \mathcal{F}_k$ in Anc-F-QI for all $k$ .*

Roughly speaking,

$$\|f_k\| \sim \|Tf_{k-1}\|_\infty = \left\| r + P\max_{a \in \mathcal{A}} f_{k-1} \right\|_\infty \lesssim R + \|f_{k-1}\|_\infty \lesssim kR + \|f_0\|_\infty,$$

so we increase the function bound as $kR$.

## 4.2 IID dataset

In this subsection, we study sample complexity with IID dataset.

**Assumption 8** (IID dataset). *There is a distribution $\mu$ such that the dataset is $D = \{s_i, a_i, r_i, s_i'\}_{i=1}^n$ generated IID with $(s_i, a_i) \sim \mu$ and $s_i' \sim P(\cdot \mid s_i, a_i)$ for $i = 1, \ldots, n$.*

Since we consider possibly infinite function space, as measurement of the capacity of function space, we use covering number [21, 83].

**Definition 2.** *An $\epsilon$-cover of set $S$ with respect to metric $d$ is a set $\{\theta_i\}_{i=1}^N \subset S$ such that for all $\theta \in S$, there is an $i \in \{1, \ldots, N\}$ such that $d(\theta, \theta^i) \leq \epsilon$. The covering number $\mathcal{N}(\epsilon; S, d)$ is the cardinality of the smallest $\epsilon$-cover. By convention, we define $\mathcal{N}(+\infty; S, d) = 1$.*

We now present lemma which bounds approximation error of Anc-F-QI for IID dataset.

**Lemma 1.** *Assume Assumptions 1, 2, 3, 7, and 8 (Bellman optimality equation, existence of argmin, star-shaped function space, increasing function range, IID dataset). Let $\mu$ be the distribution generating the dataset. Let $\epsilon > 0$ and $\delta > 0$. With probability $1 - \delta$, $\{f_k, \hat{T}f_k\}_{k=0}^{K-1}$ of Anc-F-QI with $\lambda_k = \frac{k}{k+2}$ satisfies*

$$\|Tf_k - \hat{T}f_k\|_{\mu,2}^2 \leq \frac{60(k+2)^2 R^2 \ln(2KN_{k,\epsilon}N_{k+1,\epsilon}/\delta)}{n} + 3\epsilon + 13\epsilon_B(\mathcal{F}_k, \mathcal{F}_{k+1}),$$

*where*

$$N_{k,\epsilon} = \mathcal{N}(\tfrac{\epsilon}{108(2k+1)R}; \mathcal{F}_k, \|\cdot\|_\infty), \quad for \; k = 0, 1, \ldots, K - 1.$$

We defer the proofs to Appendix D, but we quickly note that the proof is based on Bernstein inequality and is motivated by [21, 18].

Lemma 1 tells that the square of approximation error of the Bellman operator decreases sublinearly with respect to number of sample. Combining Theorem 1 and Lemma 1, we obtain following sample complexity result of Anc-F-QI with IID dataset.

**Theorem 1.** *Assume Assumptions 1, 2, 3, 5, 7, and 8 (Bellman optimality equation, existence of argmin, star-shaped function space, uniform stochastic transition, increasing function range, IID dataset). Let $\mu$ be the distribution generating the dataset. Let $\epsilon > 0$ and $\delta > 0$. With probability*

$1 - \delta$, the policy error of Anc-F-QI with $\lambda_k = \frac{k}{k+2}$ and $K = \lceil 18 C_\mu^{1/2} \|Q^{\pi_\star}\|_{2,\mu}/\epsilon \rceil$ satisfies $\|g^{\pi_\star} - g^{\pi_K}\|_\infty \le \epsilon + 3K C_\mu^{1/2} \max_{k=0,\ldots,K-1} \sqrt{\epsilon_B(\mathcal{F}_k, \mathcal{F}_{k+1})}$ with sample complexity

$$n = \tilde{\mathcal{O}}\left( \frac{R^2 C_\mu^3 \|Q^{\pi_\star}\|_{2,\mu}^4 \log(N_\epsilon^2/\delta)}{\epsilon^6} \right),$$

where $\tilde{\mathcal{O}}$ ignores all logarithmic factors except the logarithmic dependence on the covering number $N_\epsilon$ defined as

$$N_\epsilon = \max_{k=1,\ldots,K} N_{k,\epsilon}, \qquad N_{k,\epsilon} = \mathcal{N}\left( \frac{\epsilon^4}{10^6 k R C_\mu^2 \|Q^{\pi_\star}\|_{2,\mu}^2}; \mathcal{F}_k, \|\cdot\|_\infty \right), \quad \text{for } k = 1,\ldots,K.$$

*Alternatively assume Assumptions 1, 2, 3, 6, 7, and 8 (Bellman optimality equation, existence of argmin, star-shaped function space, uniform future state distribution, increasing function range, IID dataset). Let $\mu$ be the distribution generating the dataset and $\rho$ be an arbitrary distribution on $\mathcal{S} \times \mathcal{A}$. Let $\epsilon > 0$ and $\delta > 0$. With probability $1 - \delta$, the policy error of Anc-F-QI with $\lambda_k = \frac{k}{k+2}$ and $K = \lceil 18 C_{\mu,\rho}^{1/2} \|Q^{\pi_\star}\|_{2,\mu}/\epsilon \rceil$, satisfies $\|g^{\pi_\star} - g^{\pi_K}\|_{2,\rho} \le \epsilon + 3K C_{\mu,\rho}^{1/2} \max_{k=0,\ldots,K-1} \sqrt{\epsilon_B(\mathcal{F}_k, \mathcal{F}_{k+1})}$ with sample complexity*

$$n = \tilde{\mathcal{O}}\left( \frac{R^2 C_{\mu,\rho}^3 \|Q^{\pi_\star}\|_{2,\mu}^4 \log(N_\epsilon^2/\delta)}{\epsilon^6} \right),$$

*where $\tilde{\mathcal{O}}$ ignores all logarithmic factors except the logarithmic dependence on the covering number $N_\epsilon$ defined as*

$$N_\epsilon = \max_{k=1,\ldots,K} N_{k,\epsilon}, \qquad N_{k,\epsilon} = \mathcal{N}\left( \frac{\epsilon^4}{10^6 k R C_{\mu,\rho}^2 \|Q^{\pi_\star}\|_{2,\mu}^2}; \mathcal{F}_k, \|\cdot\|_\infty \right), \quad \text{for } k = 1,\ldots,K$$

In the Appendix D, we show the full sample complexity with the logarithmic factors.

Under the additional assumption of Bellman completeness ($\epsilon_B = 0$), this theorem guarantee that Anc-F-QI produces an $\epsilon$-optimal policy with $\tilde{\mathcal{O}}(1/\epsilon^6)$ sample complexity. To the best of our knowledge, this is the first sample complexity result only assuming the Bellman optimality equation or a weakly communicating MDP. In Section 5, we improve this sample complexity to $\tilde{\mathcal{O}}(1/\epsilon^4)$ using the relative normalization mechanism.

### 4.3   Single-trajectory dataset

In this subsection, we study sample complexity with single-trajectory dataset.

**Assumption 9** (single-trajectory dataset). *For given behavior policy $\pi_b$ and initial distribution $\nu$ on $\mathcal{S}$, dataset is $D = \{s_i, a_i, r_i\}_{i=1}^n$ where $s_1 \sim \nu$, $a_i \sim \pi_b(\cdot \mid s_i)$, $s_{i+1} \sim P(\cdot \mid s_i, a_i)$.*

The main technical challenge with single-trajectory data is handling the dependency between samples. Following [4, 3], we introduce the following $\beta$-mixing condition ensuring that samples are sufficiently representative and rapidly mixing.

**Definition 3** ($\beta$-mixing). *Let $\{Z_t\}_{t=1}^\infty$ be a stochastic process. Denote by $Z^{1:t}$ the collection of $(Z_1, \ldots, Z_t)$ where we allowed $t = \infty$. Let $\sigma(Z^{i:j})$ denote the $\sigma$-algebra generated by $Z^{i:j} (i \le j)$. The $m$-th $\beta$-mixing coefficient of $\{Z_t\}$ is defined as*

$$\beta_m = \sup_{t \ge 1} \mathbb{E}\left[ \sup_{B \in \sigma(Z^{t+m:\infty})} |P(B \mid Z^{1:t}) - P(B)| \right].$$

*$\{Z_t\}$ is said to be $\beta$-mixing if $\beta_m \to 0$ as $m \to \infty$. In particular, we say that a $\beta$-mixing process mixes at exponent rate with parameters $\bar{\beta}, b, \kappa > 0$ if $\beta_m \le \bar{\beta} exp(-bm^\kappa)$ holds for all $m \ge 0$.*

Roughly speaking, the $\beta$-mixing condition ensures that future samples depend weakly on the past samples. We assume that our single-trajectory is $\beta$-mixing and the distribution is in a steady state, following [4, 3].

**Assumption 10** ($\beta$-mixing single-trajectory). *For single-trajectory dataset $\{s_i, a_i, r_i\}_{i=1}^n$, assume that $s_i$ is strictly stationary with $s_i \sim \nu$ and $\beta$-mixing at exponent rate with parameters $\bar{\beta}, b, \kappa > 0$.*

Again, following [4, 3], as measurement of the capacity of function space, we use pseudo dimension which has been widely studied for complexity analyses with various function classes [2, 83].

**Definition 4** (pseudo dimension). *For a given function class $\mathcal{F}$ of binary-valued functions, we say the set $x_1^n = (x_1, \ldots, x_n)$ is shattered by $\mathcal{F}$ if cardinality of $\{(f(x_1), \ldots, f(x_n)) : f \in \mathcal{F}\}$ is $2^n$. The VC-dimension $V_{\mathcal{F}}$ of $\mathcal{F}$ is defined as the largest integer $n$ such that there exist the set $x_1^n$ shattered by $\mathcal{F}$. For a given class $\mathcal{F}$ of real-valued functions, the pseudo-dimension $V_{\mathcal{F}}$ of is defined as the VC-dimension of the set of indicator function of the subgraphs of functions in $\mathcal{F}$.*

We now present lemma which bounds approximation error of of Anc-F-QI for single-trajectory dataset.

**Lemma 2.** *Assume Assumptions 1, 2, 3, 7, 9, and 10 (Bellman optimality equation, existence of argmin, star-shaped function space, increasing function range, single-trajectory dataset, $\beta$-mixing single-trajectory). Let $\mu$ be the distribution generating the dataset defined as $\mu(s,a) = \nu(s)\pi_b(a \mid s)$. Let $\epsilon > 0$ and $\delta > 0$. With probability $1 - \delta$, $\{f_k, \hat{T}f_k\}_{k=0}^{K-1}$ of Anc-F-QI with $\lambda_k = \frac{k}{k+2}$ satisfies*

$$\|Tf_k - \hat{T}f_k\|_{\mu,2}^2 \leq \sqrt{\frac{c_{0,k}(\max\{c_{0,k}/b, 1\})^{1/\kappa}}{c_{2,k}n}} + \epsilon_B(\mathcal{F}_k, \mathcal{F}_{k+1}),$$

*where $c_{0,k} = (V_{\mathcal{F}_{k+1}} + V_{(\mathcal{F}_k)_{max}}) \log n/2 + \log(e/(K\delta)) + \log(\max(c_{1,k}, \bar{\beta})), c_{1,k} = 16e^2(V_{\mathcal{F}_{k+1}} + 1)(V_{(\mathcal{F}_k)_{max}} + 1)(24e)^{V_{\mathcal{F}_{k+1}} + V_{(\mathcal{F}_k)_{max}}}, c_{2,k} = \frac{1}{512(2k+3)^4 R^4}, V_{(\mathcal{F}_k)_{max}} = 2|\mathcal{A}|V_{\mathcal{F}_k} \log(3|\mathcal{A}|).$*

We defer the proofs to Appendix D, but we quickly note that the proof strategy closely follow [4, 3] and relies on the Hoeffding inequality under a mixing condition.

Rougbly speaking, Lemma 2 tells that the square of approximation error of the Bellman operator decreases at a $1/\sqrt{n}$ rate with respect to number of sample. Combining Theorem 1 and Lemma 2, we obtain following sample complexity result of Anc-F-QI with single-trajectory dataset.

**Theorem 2.** *Assume Assumptions 1, 2, 3, 5, 7, 9, and 10 (Bellman optimality equation, existence of argmin, star-shaped function space, uniform stochastic transition, increasing function range, single-trajectory dataset, $\beta$-mixing single-trajectory). Let $\mu$ be the distribution generating the dataset defined as $\mu(s,a) = \nu(s)\pi_b(a \mid s)$. Let $\epsilon > 0$ and $\delta > 0$. With $1 - \delta$ probability, the policy error of Anc-F-QI with $\lambda_k = \frac{k}{k+2}$ and $K = \lceil 9C_\mu^{1/2}\|Q^{\pi_\star}\|_{2,\mu}/\epsilon \rceil$ satisfies $\|g^{\pi_\star} - g^{\pi_K}\|_\infty \leq \epsilon + KC_\mu^{1/2} \max_{k=0,\ldots,K-1} \sqrt{\epsilon_B(\mathcal{F}_k, \mathcal{F}_{k+1})}$ with sample complexity*

$$n = \tilde{\mathcal{O}}\left(1/\epsilon^{12}\right),$$

*where $\tilde{\mathcal{O}}$ only shows the dependence on $\epsilon$. Alternatively, Assume Assumptions 1, 2, 3, 6, 7, 9, and 10 (Bellman optimality equation, existence of argmin, star-shaped function space, uniform future state distribution, increasing function range, single-trajectory dataset, $\beta$-mixing single-trajectory). Let $\mu$ be the distribution generating the dataset defined as $\mu(s,a) = \nu(s)\pi_b(a \mid s)$ and $\rho$ be an arbitrary distribution on $\mathcal{S} \times \mathcal{A}$. Let $\epsilon > 0$ and $\delta > 0$. With $1 - \delta$ probability, the policy error of Anc-F-QI with $\lambda_k = \frac{k}{k+2}$ and $K = \lceil 9C_{\mu,\rho}^{1/2}\|Q^{\pi_\star}\|_{2,\mu}/\epsilon \rceil$ satisfies $\|g^{\pi_\star} - g^{\pi_K}\|_{2,\rho} \leq \epsilon + KC_{\mu,\rho}^{1/2} \max_{k=0,\ldots,K-1} \sqrt{\epsilon_B(\mathcal{F}_k, \mathcal{F}_{k+1})}$ with sample complexity*

$$n = \tilde{\mathcal{O}}\left(1/\epsilon^{12}\right),$$

*where $\tilde{\mathcal{O}}$ only shows the dependence on $\epsilon$.*

In the Appendix D, we show the full sample complexity with all of the factors.

To the best of our knowledge, this is the first sample complexity result with single-trajectory data in the average-reward setup. In Section 5, we improve this sample complexity to $\tilde{\mathcal{O}}(\epsilon^{-8})$ using the relative normalization mechanism

# 5 Relative Anchored Fitted Q-Iteration

In this section, we propose *Relative Anchored Fitted Q-Iteration (R-Anc-F-QI)* and improve the sample complexity. We are motivated by the classical *relative value iteration* [90]. In the tabular case, it is known that standard VI diverges in the average-reward setup [66, Theorem 9.4.1], and relative value iteration normalizes the divergent vectors [66, Section 8.5.5]. In the case of (Anchored) Fitted Q-Iteration, this normalization allows the $f_k$ functions to be bounded and removes the inefficiency associated with the increasing function classes described in Section 4.1.

---

**Algorithm 2** Relative Anchored Fitted Q-Iteration $(D, K, \mathcal{F}, \{\lambda_i\}_{i=1}^K)$

---

**Input**: $D = \{s_i, a_i, r_i, s_i'\}_{i=1}^n$, $f_0 = 0$, $K \geq 1$, $\{\lambda_i\}_{i=1}^K \subset (0,1)$
**for** $k = 0, 1, \ldots, K-1$ **do**
$\quad \hat{T} f_k = \operatorname{argmin}_{f \in \mathcal{F}} \sum_{i=1}^n \left( f(s_i, a_i) - r_i - \max_{a \in \mathcal{A}} f_k(s_i', a) \right)^2$
$\quad f_{k+1} = (1 - \lambda_{k+1}) f_0 + \lambda_{k+1} (\hat{T} f_k - \frac{\max \hat{T} f_k + \min \hat{T} f_k}{2} \mathbf{1})$
**end for**
$\pi(a \mid s) = \operatorname{argmax}_{a \in \mathcal{A}} f_K(s, a)$
**Output** $\pi, f_K$

---

The only difference with Anchored Fitted Q-Iteration is the subtraction of $\frac{\max \hat{T} f_k + \min \hat{T} f_k}{2} \mathbf{1}$ in the second line of the for-loop. By direct calculation, we can check that $\left\| f - \frac{\max f + \min f}{2} \mathbf{1} \right\|_\infty \leq \|f\|_\infty$ and subtracting a uniform constant does not effect on greedy policy due the fact that the Bellman operator satisfies $T(c\mathbf{1} + x) = c\mathbf{1} + T(x)$. Thus, we can still apply Proposition 1 to Relative Anchored Fitted Q-Iteration.

**Assumption 11** (normalized function space). *If $f \in \mathcal{F}$, $f - \frac{\max f + \min f}{2} \mathbf{1} \in \mathcal{F}$.*

This assumption ensures that the normalization operation is well ldefined.

**Assumption 12** (range of function space). $\mathcal{F} \subset \{f : \mathcal{S} \times \mathcal{A} \to [-2 \|Q^{\pi_\star}\|_\infty, 2 \|Q^{\pi_\star}\|_\infty] \mid f \in B(S \times A)\}$, *where $Q^{\pi_\star}$ is solution of Bellman optimality equation.*

Now, unlike increasing function range used for the non-relative Anchored Fitted Q-Iteration, we now have a function space bounded by $Q^{\pi_\star}$. This difference leads to improved efficiency as the following sample complexity results show.

**Theorem 3.** *Assume Assumptions 1, 2, 3, 5, 8, 11, and 12 (Bellman optimality equation, existence of argmin, star-shaped function space, uniform stochastic transition, normalized function space, range of function space, IID dataset). Let $\mu$ be the distribution generating the dataset. Let $\epsilon > 0$ and $\delta > 0$. With probability $1 - \delta$, the policy error of R-Anc-F-QI with $\lambda_k = \frac{k}{k+2}$ and $K = \lceil 18 C_\mu^{1/2} \|Q^{\pi_\star}\|_{2,\mu} / \epsilon \rceil$ satisfies $\|g^{\pi_\star} - g^{\pi_K}\|_\infty \leq \epsilon + 3K C_\mu^{1/2} \sqrt{\epsilon_B(\mathcal{F}, \mathcal{F})}$ with sample complexity*

$$n = \tilde{\mathcal{O}} \left( \frac{(R + \|Q^{\pi_\star}\|_\infty)^2 \|Q^{\pi_\star}\|_\infty^2 C_\mu^3 \log(N_\epsilon^2 / \delta)}{\epsilon^4} \right),$$

*where $\tilde{\mathcal{O}}$ ignores all logarithmic factors except the logarithmic dependence on the covering number $N_\epsilon$ defined as*

$$N_\epsilon = \mathcal{N} \left( \frac{\epsilon^4}{10^6 C_\mu^2 (R + \|Q^{\pi_\star}\|_\infty) \|Q^{\pi_\star}\|_\infty^2}; \mathcal{F}, \|\cdot\|_\infty \right).$$

*Alternatively, assume Assumptions 1, 2, 3, 6, 8, 11, and 12 (Bellman optimality equation, existence of argmin, star-shaped function space, uniform future state distribution, normalized function space, range of function space, IID dataset) Let $\mu$ be the distribution generating the dataset and $\rho$ be an arbitrary distribution on $\mathcal{S} \times \mathcal{A}$. Let $\epsilon > 0$ and $\delta > 0$. With probability $1 - \delta$, the policy error of R-Anc-F-QI with $\lambda_k = \frac{k}{k+2}$ and $K = \lceil 18 C_{\mu,\rho}^{1/2} \|Q^{\pi_\star}\|_{2,\mu} / \epsilon \rceil$ satisfies $\|g^{\pi_\star} - g^{\pi_K}\|_{2,\rho} \leq \epsilon + 3K C_{\mu,\rho}^{1/2} \sqrt{\epsilon_B(\mathcal{F}, \mathcal{F})}$ with sample complexity*

$$n = \tilde{\mathcal{O}} \left( \frac{(R + \|Q^{\pi_\star}\|_\infty)^2 \|Q^{\pi_\star}\|_\infty^2 C_{\mu,\rho}^3 \log(N_\epsilon^2 / \delta)}{\epsilon^4} \right),$$

*where $\tilde{\mathcal{O}}$ ignores all logarithmic factors except the logarithmic dependence on the covering number $N_\epsilon$ defined as*

$$N_\epsilon = \mathcal{N}\left(\frac{\epsilon^4}{10^6 C_{\mu,\rho}^2 (R + \|Q^{\pi_\star}\|_\infty) \|Q^{\pi_\star}\|_\infty^2}; \mathcal{F}, \|\cdot\|_\infty\right).$$

**Theorem 4.** *Assume Assumptions 1, 2, 3, 5, 9, 10, 11, and 12 (Bellman optimality equation, existence of argmin, star-shaped function space, uniform stochastic transition, normalized function space, range of function space, single-trajectory dataset, $\beta$-mixing single-trajectory). Let $\mu$ be the distribution generating the dataset defined as $\mu(s,a) = \nu(s)\pi_b(a \mid s)$. Let $\epsilon > 0$ and $\delta > 0$. With probability $1 - \delta$, the policy error of Anc-F-QI with $\lambda_k = \frac{k}{k+2}$ and $K = \lceil 9 C_\mu^{1/2} \|Q^{\pi_\star}\|_{2,\mu}/\epsilon \rceil$ satisfies $\|g^{\pi_\star} - g^{\pi_K}\|_\infty \leq \epsilon + K C_\mu^{1/2} \sqrt{\epsilon_B(\mathcal{F},\mathcal{F})}$ with sample complexity*

$$n = \tilde{\mathcal{O}}\left(1/\epsilon^8\right),$$

*where $\tilde{\mathcal{O}}$ only shows the dependence on $\epsilon$. Alternatively, assume Assumptions 1, 2, 3, 6, 9, 10, 11, and 12 (Bellman optimality equation, existence of argmin, star-shaped function space, uniform future state distribution, normalized function space, range of function space, single-trajectory dataset, $\beta$-mixing single-trajectory). Let $\mu$ be the distribution generating the dataset defined as $\mu(s,a) = \nu(s)\pi_b(a \mid s)$ and $\rho$ be an arbitrary distribution on $\mathcal{S} \times \mathcal{A}$. Let $\epsilon > 0$ and $\delta > 0$. With probability $1 - \delta$, the policy error of Anc-F-QI with $\lambda_k = \frac{k}{k+2}$ and $K = \lceil 9 C_{\mu,\rho}^{1/2} \|Q^{\pi_\star}\|_{2,\mu}/\epsilon \rceil$ satisfies $\|g^{\pi_\star} - g^{\pi_K}\|_{2,\rho} \leq \epsilon + K C_{\mu,\rho}^{1/2} \sqrt{\epsilon_B(\mathcal{F},\mathcal{F})}$ with sample complexity*

$$n = \tilde{\mathcal{O}}\left(1/\epsilon^8\right),$$

*where $\tilde{\mathcal{O}}$ only shows the dependence on $\epsilon$.*

Indeed, with the relative normalization mechanism, we improve the sample complexities from $\tilde{\mathcal{O}}(1/\epsilon^6)$ to $\tilde{\mathcal{O}}(1/\epsilon^4)$ and $\tilde{\mathcal{O}}(1/\epsilon^{12})$ to $\tilde{\mathcal{O}}(1/\epsilon^8)$ for IID and single-trajectory data cases, respectively,

## 6 Conclusion

In this work, we introduced Anchored Fitted Q-Iteration (Anc-F-QI) and established new sample complexity results for the average-reward offline RL with general function approximation under the assumption of weakly communicating MDPs. Our approach combines the classical Fitted Q-Iteration with an anchoring mechanism, and the anchor mechanism is the crucial component that enables the finite-time analysis. Roughly speaking, we establish a $\tilde{\mathcal{O}}(1/\epsilon^6)$ sample complexity with IID data and $\tilde{\mathcal{O}}(1/\epsilon^{12})$ sample complexity with single-trajectory data. Then, using the relative normalization technique, we improve the sample complexity to $\tilde{\mathcal{O}}(1/\epsilon^4)$ and $\tilde{\mathcal{O}}(1/\epsilon^8)$ for IID and single-trajectory data, respectively.

One limitation of this work is the reliance on *full* coverage coefficients as described in Assumptions 5 and 6. Some prior work, such as [64] and [30], utilizes a weaker assumption that we refer to as *partial* coverage coefficients, albeit with much stronger structural assumptions on the MDP. Extending our analysis to relax the full coverage coefficient would be a worthwhile direction of future work. Another possible direction of future work is to utilize variance reduction techniques in the style of [84, 75, 48] to further improve the sample complexity.

## Acknowledgments and Disclosure of Funding

This work is supported by the National Research Foundation of Korea (NRF) grant funded by the Korean government (No.RS-2024-00421203).

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

# A Prior works

**Average-Reward MDP**   The setup of average reward MDPs was introduced in the dynamic programming literature by [36], and [9] established a theoretical framework for their analysis. In reinforcement learning (RL), average-reward MDP was mainly considered in the sample-based setup where the transition matrix and reward are unknown [57, 23]. For this setup, various methods were proposed: model-based methods [41, 97], Q-learning methods [89, 85], and policy gradient methods [5, 62, 46]. Sample complexity to obtain $\epsilon$-optimal under generative model [86, 96, 48, 54, 40] and for regret minimization [15, 38, 95, 11] also have been actively studied.

**Value Iteration**   Value iteration (VI) was first introduced in the dynamic programming literature [6] and serve as a fundamental algorithm to compute the value functions. The sample-based variants, such as TD-Learning [77], Fitted Value Iteration [25, 61], and Deep Q-Network [58] are the workhorses of modern reinforcement learning algorithms [8, 78, 79]. VI is also routinely applied in diverse settings, including factored MDPs [68], robust MDPs [47], MDPs with reward machines [12], MDPs with options [29], and generative model [84, 75, 48].

The convergence of VI in average-reward MDPs also has been extensively studied. For unichain MDPs, delta coefficient, ergodicity coefficient, and the J-stage span contraction demonstrate the linear rate of VI [74, 37, 27, 81]. When MDP is multichain, it is known that policy error of VI might not converge to zero [22, Example 4]. Even with the aperiodicity assumption, VI guarantees only asymptotic convergence. [66, Theorem 9.4.5]. [72, 73] established necessary and sufficient conditions of convergence of VI and asymptotic linear convergence on Bellman error.

**Offline Reinforcement Learning**   In offline RL, the agent learns decision-making strategies utilizing precollected data [53]. This framework is often applied when interaction with the environment can be expensive, and the quantities of data that can be gathered online are substantially lower than the precollected dataset [19, 43, 53]. Consequently, various offline RL methods have been actively proposed [25, 76, 45, 1], and Fitted Q-Iteration is one of the representative methods based on sample-based value iteration with function approximation [25, 61].

One issue in offline RL is the distribution mismatch between the behavior policy that collected the data and the learned policy of the agent [44, 87]. For theoretical analysis, *coverage coefficient* is assumed to ensure that offline dataset sufficiently explores whole state and action space. [60, 71, 80]. Under this assumption, sample complexity of offline RL methods actively analyzed [4, 69, 18, 64], and in particular, an $L_p$ bound of approximate value iteration was obtained, which in turn yields convergence results for Fitted Q-Iteration [60, 61]. More recently, several works succeeded relaxing the full coverage assumption to partcal coverage [56, 67, 91, 42].

Another issue in offline RL is the representation capacity of the chosen function space. To handle large state space and action spaces, many RL frameworks including offline RL use function approximation, ranging from linear functions [24] and nonlinear (general) functions such as neural networks [26] and kernel functions [17]. In offline RL, the *inherent Bellman error* measures the approximation error incurred when projecting the output of Bellman operator into chosen function space, and Bellamn completeness assumes the inherent Bellman error is zero [61, 18]. Most sample complexity analyses in offline RL rely on inherent Bellman error or Bellman completeness assumption [56, 67, 91, 42]. Recently, however, several works achieved finite sample complexity under weaker realizabiltiy assumption, which only requires that optimal function value lies within chosen function space [92, 94].

Most of prior works in offline RL focused on discounted-reward setup, and to the best of our knowledge, two prior works established the finite sample complexity in the offline average-reward setup [63, 30]. Both proposed a primal-dual approach, reformulating the Bellman equation as a bilinear saddle-point problem, to obtain an $\epsilon$-optimal policy under partial coverage. However, they imposed restrictive structural assumptions on MDP such as uniform mixing or linearity and considered only IID dataset. (See the Table 1.)

# B Preliminaries

The followings are inequalities from prior works used in the proof.

**Fact 1** (Bernstein inequality). *Let $X_1, \ldots, X_n$ are indepedent random variables. If $X_i \leq b$ for all $i$, then*

$$\mathbb{P}\left(\frac{1}{n}\sum_{i=1}^{n} X_i - \mathbb{E}[X_i] \geq \epsilon\right) \leq exp\left[-\frac{n^2\epsilon^2}{2\sum_{i=1}^{n}\mathbb{E}[X_i^2] + nb\epsilon/3}\right]$$

*Furthermore, if all the $\mathbb{E}[X_i^2]$ are equal, with $1 - \delta$ probability,*

$$\frac{1}{n}\sum_{i=1}^{n} X_i - \mathbb{E}X_i \leq \sqrt{2\mathbb{E}[X_1^2]\ln(1/\delta)/n} + \frac{2b\ln(1/\delta)}{3n}.$$

**Fact 2** ([4], Lemma 4). *Suppose that $Z_1, \ldots, Z_n \in \mathcal{Z}$ is a stationary $\beta$-mixing process with mixing coefficients $\beta_m$, $Z_t' \in \mathcal{Z}(t \in H)$ are the block-independent ghost samples. $H = \{2ik_N + j : 0 \leq i < m_n, 1 \leq j \leq k_N\}$ and $\mathcal{F}$ is permissible class of $\mathcal{Z} \to [-M, M]$ functions. Then*

$$P\left(\sup_{f\in\mathcal{F}}\left|\frac{1}{N}\sum_{n=1}^{N} f(Z_n) - \mathbb{E}[f(Z_1)]\right| > \epsilon\right) \leq 16\mathbb{E}[\mathcal{N}(\epsilon/8, \mathcal{F}, l_{(Z_t')_{t\in H}})]e^{-\frac{m_N\epsilon^2}{128M^2}} + 2m_N\beta_{k_N+1}.$$

# C Omitted proofs in Section 3

## C.1 Proof of Proposition 1

Define the limiting matrix $\mathcal{P}_*^{\pi}$ as the Cesàro limit of $\mathcal{P}^{\pi}$, i.e., $\mathcal{P}_*^{\pi} = \lim\frac{1}{n}\sum_{i=1}^{n}(\mathcal{P}^{\pi})^i$. (The limiting matrix always exists for finite state-action spaces [66, Appendix A.4].) Then, $\mathcal{P}_*^{\pi}$ is stochastic and, by definition, $g^{\pi} = \mathcal{P}_*^{\pi}r$ [66, Proposition 8.1.1].

We first prove following lemma.

**Lemma 3.** *Let $\lambda_{K+1} = 1$. Under Assumption 1 (Bellman optimality equation), the policy error of Apx-Anc-QI satisfies*

$$g^{\pi_\star} - g^{\pi_K} = \mathcal{P}_*^{\pi_K}(g^{\pi_\star} - TQ^K + Q^K)$$

$$\leq \mathcal{P}_*^{\pi_K}\left(\sum_{l=0}^{K}\Pi_{i=l+1}^{K}\lambda_i(\lambda_{l+1} - \lambda_l)\Pi_{i=l+1}^{K}\mathcal{P}^{\pi_i}\left(\sum_{m=0}^{l}\Pi_{i=m+1}^{l}\lambda_i(1 - \lambda_m)(\mathcal{P}^{\pi_\star})^{l+1-m} - I\right)(Q^0 - Q^{\pi_\star})\right.$$

$$\left. + \sum_{l=1}^{K}\Pi_{i=l}^{K}\lambda_i\left(\sum_{m=l}^{K}(\lambda_{m+1} - \lambda_m)\Pi_{i=m+1}^{K}\mathcal{P}^{\pi_i}(\mathcal{P}^{\pi_\star})^{m+1-l} + \Pi_{i=l+1}^{K}\mathcal{P}^{\pi_i}(\lambda_l\mathcal{P}^{\pi_l} - I)\right)\epsilon_l\right).$$

*Proof of Lemma 3.* By definition of Apx-Anc-QI, we have

$$TQ^K - Q^K$$
$$= (1 - \lambda_K)(TQ^K - Q^0) + \lambda_K(TQ^K - TQ^{K-1}) - \lambda_K\epsilon_K$$
$$\geq (1 - \lambda_K)(TQ^K - Q^0) + \lambda_K\mathcal{P}^{\pi_K}(Q^K - Q^{K-1}) - \lambda_K\epsilon_K$$
$$\geq (1 - \lambda_K)(TQ^K - Q^0) - \lambda_K\epsilon_K$$
$$+ \lambda_K\mathcal{P}^{\pi_K}((\lambda_K - \lambda_{K-1})(TQ^{K-1} - Q^0) + \lambda_{K-1}(TQ^{K-1} - TQ^{K-2}) + \lambda_K\epsilon_K - \lambda_{K-1}\epsilon_{K-1})$$
$$\geq \sum_{l=0}^{K}\Pi_{i=l+1}^{K}\lambda_i(\lambda_{l+1} - \lambda_l)\Pi_{i=l+1}^{K}\mathcal{P}^{\pi_i}(TQ^l - Q^0) + \sum_{l=1}^{K}\Pi_{i=l}^{K}\lambda_i\Pi_{i=l+1}^{K}\mathcal{P}^{\pi_i}(\lambda_l\mathcal{P}^{\pi_l} - I)\epsilon_l$$

where first inequality comes from greedy policy and last inequality comes from induction.

For any $0 \leq l \leq K$,

$$TQ^l - Q^0$$
$$= TQ^l - Q^{\pi_\star} - (Q^0 - Q^{\pi_\star})$$
$$= TQ^l - TQ^{\pi_\star} + g^{\pi_\star} - (Q^0 - Q^{\pi_\star})$$
$$\geq \mathcal{P}^{\pi_\star}(Q^l - Q^{\pi_\star}) + g^{\pi_\star} - (Q^0 - Q^{\pi_\star})$$
$$= \mathcal{P}^{\pi_\star}(\lambda_l(TQ^{l-1} - Q^{\pi_\star}) + (1 - \lambda_l)(Q^0 - Q^{\pi_\star}) + \lambda_l \epsilon_l) + g^{\pi_\star} - (Q^0 - Q^{\pi_\star})$$
$$\geq \left( \sum_{m=0}^{l} \Pi_{i=m+1}^{l} \lambda_i (\mathcal{P}^{\pi_\star})^{l+1-m}(1 - \lambda_m) - I \right)(Q^0 - Q^{\pi_\star}) + \sum_{m=0}^{l} \Pi_{i=m+1}^{l} \lambda_i g^{\pi_\star}$$
$$+ \sum_{m=1}^{l} \Pi_{i=m}^{l} \lambda_i (\mathcal{P}^{\pi_\star})^{l+1-m} \epsilon_m,$$

where second equality comes from Bellman optimality equation. By combining previous two inequalities, we get

$$TQ^K - Q^K$$
$$\geq \sum_{l=0}^{K} \Pi_{i=l+1}^{K} \lambda_i (\lambda_{l+1} - \lambda_l) \Pi_{i=l+1}^{K} \mathcal{P}^{\pi_i} \sum_{m=0}^{l} \Pi_{i=m+1}^{l} \lambda_i g^{\pi_\star}$$
$$+ \sum_{l=0}^{K} \Pi_{i=l+1}^{K} \lambda_i (\lambda_{l+1} - \lambda_l) \Pi_{i=l+1}^{k} \mathcal{P}^{\pi_i} \left( \sum_{m=0}^{l} \Pi_{i=m+1}^{l} \lambda_i (\mathcal{P}^{\pi_\star})^{l+1-m}(1 - \lambda_m) - I \right)(Q^0 - Q^{\pi_\star})$$
$$+ \sum_{l=1}^{K} \Pi_{i=l}^{K} \lambda_i \Pi_{i=l+1}^{K} \mathcal{P}^{\pi_i}(\lambda_l \mathcal{P}^{\pi_l} - I)\epsilon_l$$
$$+ \sum_{l=1}^{K} \sum_{m=1}^{l} \Pi_{i=l+1}^{K} \lambda_i (\lambda_{l+1} - \lambda_l) \Pi_{i=l+1}^{K} \mathcal{P}^{\pi_i} \Pi_{i=m}^{l} \lambda_i (\mathcal{P}^{\pi_\star})^{l+1-m} \epsilon_m$$
$$= g^{\pi_\star} + \sum_{l=1}^{K} \left( \sum_{m=l}^{k} \Pi_{i=m+1}^{K} \lambda_i (\lambda_{m+1} - \lambda_m) \Pi_{i=l}^{m} \lambda_i \Pi_{i=m+1}^{K} \mathcal{P}^{\pi_i} (\mathcal{P}^{\pi_\star})^{m+1-l} \right.$$
$$+ \Pi_{i=l}^{K} \lambda_i \Pi_{i=l+1}^{K} \mathcal{P}^{\pi_i}(\lambda_l \mathcal{P}^{\pi_l} - I) \bigg) \epsilon_l$$
$$+ \sum_{l=0}^{K} \Pi_{i=l+1}^{K} \lambda_i (\lambda_{l+1} - \lambda_l) \Pi_{i=l+1}^{K} \mathcal{P}^{\pi_i} \left( \sum_{m=0}^{l} \Pi_{i=m+1}^{l} \lambda_i (1 - \lambda_m)(\mathcal{P}^{\pi_\star})^{l+1-m} - I \right)(Q^0 - Q^{\pi_\star}).$$

This implies

$$TQ^K - Q^K - g^{\pi_\star}$$
$$\geq \sum_{l=0}^{K} \Pi_{i=l+1}^{K} \lambda_i (\lambda_{l+1} - \lambda_l) \Pi_{i=l+1}^{K} \mathcal{P}^{\pi_i} \left( \sum_{m=0}^{l} \Pi_{i=m+1}^{l} \lambda_i (1 - \lambda_m)(\mathcal{P}^{\pi_\star})^{l+1-m} - I \right)(Q^0 - Q^{\pi_\star})$$
$$+ \sum_{l=1}^{K} \Pi_{i=l}^{K} \lambda_i \left( \sum_{m=l}^{K} (\lambda_{m+1} - \lambda_m) \Pi_{i=m+1}^{K} \mathcal{P}^{\pi_i} (\mathcal{P}^{\pi_\star})^{m+1-l} + \Pi_{i=l+1}^{K} \mathcal{P}^{\pi_i}(\lambda_l \mathcal{P}^{\pi_l} - I) \right) \epsilon_l.$$

Finally, following the proof of [66, Theorem 8.5.5], we have

$$g^{\pi_\star} - g^{\pi_K} = \mathcal{P}_\ast^{\pi_K}(g^{\pi_\star} - r) = \mathcal{P}_\ast^{\pi_K}(g^{\pi_\star} - r - \mathcal{P}^{\pi_K} Q^K + Q^K)$$
$$= \mathcal{P}_\ast^{\pi_K}(g^{\pi_\star} - TQ^K + Q^K),$$

where first equality comes from Bellman optimality equation and second equality comes from property of limiting matrix. This implies that

$$g^{\pi_\star} - g^{\pi_K} = \mathcal{P}_*^{\pi_K}(g^{\pi_\star} - TQ^K + Q^K)$$

$$\leq \mathcal{P}_*^{\pi_K}\bigg(\sum_{l=0}^{K} \Pi_{i=l+1}^{K}\lambda_i(\lambda_{l+1} - \lambda_l)\Pi_{i=l+1}^{K}\mathcal{P}^{\pi_i}\bigg(\sum_{m=0}^{l}\Pi_{i=m+1}^{l}\lambda_i(1-\lambda_m)(\mathcal{P}^{\pi_\star})^{l+1-m} - I\bigg)(Q^0 - Q^{\pi_\star})$$

$$+ \sum_{l=1}^{K}\Pi_{i=l}^{K}\lambda_i\bigg(\sum_{m=l}^{K}(\lambda_{m+1}-\lambda_m)\Pi_{i=m+1}^{K}\mathcal{P}^{\pi_i}(\mathcal{P}^{\pi_\star})^{m+1-l} + \Pi_{i=l+1}^{K}\mathcal{P}^{\pi_i}(\lambda_l\mathcal{P}^{\pi_l} - I)\bigg)\epsilon_l\bigg).$$

$\square$

The following are lemmas about coverage coefficient $C_{\mu,\rho}$.

**Lemma 4.** *If $\mathcal{P}_1$ and $\mathcal{P}_2$ are stochastic matrix satisfying $\rho^\top\mathcal{P}_i \leq C_{\mu,\rho}\mu$ for $i = 1, 2$ and given distribution $\mu$ and $\rho$ on $\mathcal{S}\times\mathcal{A}$, then $\rho^\top(a\mathcal{P}_1 + (1-a)\mathcal{P}_2) \leq C_{\mu,\rho}\mu$ for $0 \leq a \leq 1$.*

**Lemma 5.** *Under Assumption 6 (uniform future state distribution),*

$$\sup_{\pi_1,\pi_2,\ldots\pi_k}\bigg\|\frac{\rho^\top\mathcal{P}_*^{\pi_\star}\mathcal{P}^{\pi_1}\mathcal{P}^{\pi_2}\cdots\mathcal{P}^{\pi_k}(\cdot)}{\mu(\cdot)}\bigg\|_\infty \leq C_{\mu,\rho}$$

*where $\pi_\star\pi_1, \pi_2, \ldots\pi_k$ represents an arbitrary sequence of policies with optimal policy.*

*Proof.* Under Assumption 6, for any non negative integer $n$, we have $\rho^\top(\mathcal{P}^{\pi_\star})^n\mathcal{P}^{\pi_1}\mathcal{P}^{\pi_2}\cdots\mathcal{P}^{\pi_k}(\cdot) \leq C_{\mu,\rho}\mu$. This implies $\rho^\top\mathcal{P}_*^{\pi_\star}\mathcal{P}^{\pi_1}\mathcal{P}^{\pi_2}\cdots\mathcal{P}^{\pi_k}(\cdot) \leq C_{\mu,\rho}\mu$ by definition of limiting matrix. $\square$

**Lemma 6.** *If $\mathcal{P}$ is stochastic matrix satisfying $\rho^\top\mathcal{P} \leq C_{\mu,\rho}\mu^\top$ for given distribution $\mu$ and $\rho$ on $\mathcal{S}\times\mathcal{A}$, then $\|\mathcal{P}Q\|_{p,\rho} \leq C_\mu^{1/p}\|Q\|_{p,\mu}$.*

*Proof.* Since $|\mathcal{P}Q(s,a)|^p = |\mathbb{E}_{(s',a')\sim\mathcal{P}(\cdot\,|\,s,a)}[Q(s',a')]|^p \leq \mathbb{E}_{(s',a')\sim\mathcal{P}(\cdot\,|\,s,a)}[|Q(s',a')|^p]) = \mathcal{P}|Q|^p(s,a)$ by Jensen's inequality, $\rho^\top|\mathcal{P}Q|^p \leq \rho^\top\mathcal{P}|Q|^p \leq C_{\mu,\rho}\mu^\top|Q|^p$. $\square$

Now, we are ready to prove Proposition 1.

*Proof of Proposition 1.* By Lemma 3,

$$g^{\pi_\star} - g^{\pi_K}$$

$$\leq \mathcal{P}_*^{\pi_K}\bigg(\sum_{l=0}^{K} \Pi_{i=l+1}^{K}\lambda_i(\lambda_{l+1} - \lambda_l)\Pi_{i=l+1}^{K}\mathcal{P}^{\pi_i}\bigg(\sum_{m=0}^{l}\Pi_{i=m+1}^{l}\lambda_i(1-\lambda_m)(\mathcal{P}^{\pi_\star})^{l+1-m} - I\bigg)(Q^0 - Q^{\pi_\star})$$

$$+ \sum_{l=1}^{K}\Pi_{i=l}^{K}\lambda_i\bigg(\sum_{m=l}^{K}(\lambda_{m+1}-\lambda_m)\Pi_{i=m+1}^{K}\mathcal{P}^{\pi_i}(\mathcal{P}^{\pi_\star})^{m+1-l} + \Pi_{i=l+1}^{K}\mathcal{P}^{\pi_i}(\lambda_l\mathcal{P}^{\pi_l} - I)\bigg)\epsilon_l\bigg)$$

$$\leq \mathcal{P}_*^{\pi_K}\bigg(\sum_{l=0}^{K} \Pi_{i=l+1}^{K}\lambda_i(\lambda_{l+1} - \lambda_l)\Pi_{i=l+1}^{K}\mathcal{P}^{\pi_i}\bigg(\sum_{m=0}^{l}\Pi_{i=m+1}^{l}\lambda_i(1-\lambda_m)(\mathcal{P}^{\pi_\star})^{l+1-m} + I\bigg)|Q^0 - Q^{\pi_\star}|$$

$$+ \sum_{l=1}^{K}\Pi_{i=l}^{K}\lambda_i\bigg(\sum_{m=l}^{K}(\lambda_{m+1}-\lambda_m)\Pi_{i=m+1}^{K}\mathcal{P}^{\pi_i}(\mathcal{P}^{\pi_\star})^{m+1-l} + \Pi_{i=l+1}^{K}\mathcal{P}^{\pi_i}(\lambda_l\mathcal{P}^{\pi_l} + I)\bigg)|\epsilon_l|\bigg).$$

Let $\mathcal{P}_l^Q = \mathcal{P}_*^{\pi_K}\Pi_{i=l+1}^{K}\mathcal{P}^{\pi_i}\big(\sum_{m=0}^{l}\Pi_{i=m+1}^{l}\lambda_i(1-\lambda_m)(\mathcal{P}^{\pi_\star})^{l+1-m} + I\big)/2$ and $\mathcal{P}_l^\epsilon = \mathcal{P}_*^{\pi_K}\sum_{m=l}^{K}(\lambda_{m+1}-\lambda_m)\Pi_{i=m+1}^{K}\mathcal{P}^{\pi_i}(\mathcal{P}^{\pi_\star})^{m+1-l} + \Pi_{i=l+1}^{K}\mathcal{P}^{\pi_i}(\lambda_l\mathcal{P}^{\pi_l} + I)/2$. Then $\mathcal{P}_l^Q$ and $\mathcal{P}_l^\epsilon$ satisfying $\rho^\top\mathcal{P}_l^Q \leq C_{\mu,\rho}\mu$ and $\rho^\top\mathcal{P}_l^\epsilon \leq C_{\mu,\rho}\mu$ for all $0 \leq l \leq K$ by Lemma 4 and 5. Thus, we

have

$$\|g^{\pi_\star} - g^{\pi_K}\|_{p,\rho} \le 2\sum_{l=0}^{K} \Pi_{i=l+1}^{K}\lambda_i(\lambda_{l+1}-\lambda_l)\|\mathcal{P}_l|Q^0 - Q^{\pi_\star}|\|_{p,\rho} + 2\sum_{l=1}^{K}\Pi_{i=l}^{K}\lambda_i\|\mathcal{P}_l^\epsilon|\epsilon_l|\|_{p,\rho}$$

$$\le 2C_\mu^{1/p}\sum_{l=0}^{K}\Pi_{i=l+1}^{K}\lambda_i(\lambda_{l+1}-\lambda_l)\|Q^0 - Q^{\pi_\star}\|_{p,\mu} + 2C_\mu^{1/p}\sum_{l=1}^{K}\Pi_{i=l}^{K}\lambda_i\|\epsilon_l\|_{p,\mu},$$

where last inequality comes from Lemma 6. By plugging $\lambda_k = \frac{k}{k+2}$, we conclude. Note that since $C_\mu \le C_{\mu,\rho}$ for any distribution $\rho$, then choosing $\rho$ to be a Dirac distribution at each state proves the case of Assumption 5 which implies first inequality of Proposition 1. $\qquad\square$

## D  Omitted proofs in Section 4

### D.1  Proof of Lemma 1

*Proof of Lemma 1.* Let $\mathcal{F} \subset \{f : \mathcal{S} \times \mathcal{A} \to [-f_{max}, f_{max}] \mid f \in B(\mathcal{S} \times \mathcal{A})\}$ and $\mathcal{G} \subset \{f : \mathcal{S} \times \mathcal{A} \to [-g_{max}, g_{max}] \mid f \in B(\mathcal{S} \times \mathcal{A})\}$. Let $f_1, \dots, f_N$ cover the $\mathcal{F}$ and $g_1, \dots, g_{N'}$ cover the $\mathcal{G}$ where $N = \mathcal{N}(\epsilon/M; \mathcal{F}, \|\cdot\|_\infty)$, $N' = \mathcal{N}(\epsilon/M; \mathcal{G}, \|\cdot\|_\infty)$, $M = 108(R + 2f_{max})$. $\mathcal{F} \times \mathcal{G} = \cup S_{i,j}$ where $S_{i,j} = \{(f,g) : \|f - f_i\|_\infty \le \epsilon, \|g - g_j\|_\infty \le \epsilon\}$. Without loss of generality, suppose $g_{max} \le f_{max}$.

First note that $\mathbb{E}_{s_i' \sim P(\cdot \mid s_i, a_i)}[r(s_i, a_i) + \max_a g(s_i', a)] = Tg(s_i, a_i)$, $|r_i + \max_a g(s,a)| \le R + f_{max}$, $|Tg(s,a)| \le R + f_{max}$.

For arbitrary $f \in \mathcal{F}$, $g \in \mathcal{G}$, define $X_i^{f,g} = (f(s_i, a_i) - r(s_i, a_i) - \max_a g(s_i', a))^2 - (Tg(s_i, a_i) - r(s_i, a_i) - \max_a g(s_i', a))^2$. Then, $\mathbb{E}_{s_i, a_i \sim \mu, s_i' \sim P(\cdot \mid s_i, a_i)}[X_i^{f,g}] = \|Tg - f\|_{\mu,2}^2$ and $\mathbb{E}[(X_i^{f,g})^2] \le 9(R + 2f_{max})^2\|Tg - f\|_{\mu,2}^2$ since $X_i^{f,g} = (f(s_i, a_i) - Tg(s_i, a_i))(f(s_i, a_i) + Tg(s_i, a_i) - 2r(s_i, a_i) - 2\max_a g(s_i', a))$, and $|X_i^{f,g}| \le 3(R + 2f_{max})^2$.

By Bernstein inequality and union bound, with $1 - \delta$ probability, for all $\{f_i, g_j\}_{i=1,\dots,N, j=1,\dots,N'}$,

$$\|Tg_j - f_i\|_{\mu,2}^2 - \sum_{i=1}^{n} X_i^{f_i,g_j}/n \le \sqrt{\frac{18(R + 2f_{max})^2\|Tg_j - f_i\|_{\mu,2}^2 \ln(\mathcal{N}_{\mathcal{F},\mathcal{G}}/\delta)}{n}}$$

$$+ \frac{2(R + 2f_{max})^2 \ln(\mathcal{N}_{\mathcal{F},\mathcal{G}}/\delta)}{n}$$

where $\mathcal{N}_{\mathcal{F},\mathcal{G}} = \mathcal{N}(\epsilon/M; \mathcal{G}, \|\cdot\|_\infty)\mathcal{N}(\epsilon/M; \mathcal{F}, \|\cdot\|_\infty)$. Through $2\sqrt{ab} \le a + b$, we have

$$\forall f_i \in \mathcal{F}, \forall g_i \in \mathcal{G}, \quad \|Tg_j - f_i\|_{\mu,2}^2 - 2\sum_{i=1}^{n} X_i^{f_i,g_j}/n \le \frac{22(R + 2f_{max})^2 \ln(\mathcal{N}_{\mathcal{F},\mathcal{G}}/\delta)}{n}$$

Now, for covering number argument, we use following Lemma.

**Lemma 7.** *For $f \in \mathcal{F}$, $g \in \mathcal{G}$, $c > 0$, $\|Tg - f\|_{\mu,2}^2 - c\sum_{i=1}^{n} X_i^{f,g}/n$ is $(2 + 8c)(2f_{max} + R)$-Lipchitz.*

*Proof.* Since $\|Tg_1 - f_1\|_{\mu,2}^2 - \|Tg_2 - f_2\|_{\mu,2}^2 \le \mathbb{E}|(Tg_1 - Tg_2 + f_2 - f_1)(Tg_1 + Tg_2 - f_2 - f_1)| \le (\|g_1 - g_2\|_\infty + \|f_1 - f_2\|_\infty)2(R + 2f_{max})$, $\|Tg - f\|_{\mu,2}^2$ is $2(R + 2f_{max})$- Lipchitz. Also, since $|\sum_{i=1}^{n} X_i^{f_1,g_1}/n - \sum_{i=1}^{n} X_i^{f_2,g_2}/n| = \frac{1}{n}\sum_{i=1}^{n} |(\max g_2 - \max g_1 + f_1 - f_2)(f_2 + f_1 - \max g_1 - \max g_2 - 2r) - (Tg_1 - Tg_2 + \max g_2 - \max g_1)(Tg_1 + Tg_2 + \max g_2 + \max g_1 - 2r)| \le (\|g_1 - g_2\|_\infty + \|f_1 - f_2\|_\infty)2(R + 2f_{max}) + 8\|g_1 - g_2\|_\infty(f_{max} + R) \le (\|g_1 - g_2\|_\infty + \|f_1 - f_2\|_\infty)8(2f_{max} + R)$, $\sum_{i=1}^{n} X_i^{f_1,g_1}/n$ $8(2f_{max} + R)$-Lipchitz. By adding two Lipchitz functions, we obtain desired result. $\qquad\square$

By Lipchitzness of $\|Tg_j - f_i\|_{\mu,2}^2 - 2\sum_{i=1}^{n} X_i^{f_i,g_j}/n$ and definition of covering number, if $f, g \in S_{i,j}$

$$\|Tg - f\|_{\mu,2}^2 - 2\sum_{i=1}^{n} X_i^{f,g}/n - (\|Tg_j - f_i\|_{\mu,2}^2 - 2\sum_{i=1}^{n} X_i^{f_i,g_j}/n) \le \epsilon.$$

This implies that with $1 - \delta$ probability,

$$\forall f \in \mathcal{F}, \forall g \in \mathcal{G} \quad \|Tg - f\|_{\mu,2}^2 \le \epsilon + \frac{22(R + 2f_{max})^2 \ln(\mathcal{N}_{\mathcal{F},\mathcal{G}}/\delta)}{n} + 2\sum_{i=1}^{n} X_i^{f,g}/n. \quad (1)$$

By other side of Bernstein's inequality and covering number, for all $\{f_i, g_j\}_{i=1,\dots,N,j=1,\dots,N'}$, we have

$$\sum_{i=1}^{n} X_i^{f_i,g_j}/n - \|Tg_j - f_i\|_{\mu,2}^2 \le \sqrt{\frac{18(R + 2f_{max})^2 \|Tg_j - f_i\|_{\mu,2}^2 \ln(\mathcal{N}_{\mathcal{F},\mathcal{G}}/\delta)}{n}}$$
$$+ \frac{2(R + 2f_{max})^2 \ln(\mathcal{N}_{\mathcal{F},\mathcal{G}}/\delta)}{n}.$$

If $\sum_{i=1}^{n} X_i^{f_i,g_j}/n \ge \frac{4(R+2f_{max})^2 \ln(\mathcal{N}_{\mathcal{F},\mathcal{G}}/\delta)}{n}$, with $1 - \delta$ probability, for all $\{f_i, g_j\}_{i=1,\dots,N,j=1,\dots,N'}$, we have

$$\sum_{i=1}^{n} X_i^{f_i,g_j}/n - \|Tg_j - f_i\|_{\mu,2}^2 \le \sqrt{4.5 \sum_{i=1}^{n} X_i^{f_i,g_j}/n \|Tg_j - f_i\|_{\mu,2}^2} + \frac{2(R + 2f_{max})^2 \ln(\mathcal{N}_{\mathcal{F},\mathcal{G}}/\delta)}{n}$$

and by $2\sqrt{ab} \le a + b$, this implies

$$\sum_{i=1}^{n} X_i^{f_i,g_j}/n - 6.5\|Tg_j - f_i\|_{\mu,2}^2 \le \frac{4(R + 2f_{max})^2 \ln(\mathcal{N}_{\mathcal{F},\mathcal{G}}/\delta)}{n}.$$

Even if $\sum_{i=1}^{n} X_i^{f_i,g_j}/n \le \frac{4(R+2f_{max})^2 \ln(\mathcal{N}_{\mathcal{F},\mathcal{G}}/\delta)}{n}$, previous inequality still holds. Since $\sum_{i=1}^{n} X_i^{f_i,g_j}/n - 6.5\|Tg_j - f_i\|_{\mu,2}^2$ is $54(R + 2f_{max})$-Lipshitz, with similar argument, we have

$$\forall f \in \mathcal{F}, g \in \mathcal{G}, \quad \sum_{i=1}^{n} X_i^{f,g}/n - 6.5\|Tg - f\|_{\mu,2}^2 \le \epsilon + \frac{4(R + 2f_{max})^2 \ln(\mathcal{N}_{\mathcal{F},\mathcal{G}}/\delta)}{n}. \quad (2)$$

Let $\tilde{T}g = \arg\min_{f \in \mathcal{F}} \|f - Tg\|_{2,\mu}$ and $f = \tilde{T}g$ in inequality (2). Then, by definition of Inherent Bellman error,

$$\forall g \in \mathcal{G}, \quad \sum_{i=1}^{n} X_i^{\tilde{T}g,g}/n \le \epsilon + 6.5\epsilon_B + \frac{4(R + 2f_{max})^2 \ln(\mathcal{N}_{\mathcal{F},\mathcal{G}}/\delta)}{n}.$$

Also, let $f = \hat{T}g$ in inequality inequality (1). Then, by definition of $\hat{T}g$, we have $\sum_{i=1}^{n} X_i^{\hat{T}g,g} \le \sum_{i=1}^{n} X_i^{\tilde{T}g,g}$. Combining with previous inequality, with $1 - 2\delta$ probability,

$$\forall g \in \mathcal{G}, \quad \|Tg - \hat{T}g\|_{\mu,2}^2 \le 3\epsilon + 13\epsilon_B + \frac{30(R + 2f_{max})^2 \ln(\mathcal{N}_{\mathcal{F},\mathcal{G}}/\delta)}{n}.$$

Finally, let $\mathcal{G} = \mathcal{F}_k, \mathcal{F} = \mathcal{F}_{k+1}$, and $g = f_k$, and by manipulating $\delta$, we get desired result. $\square$

### D.2   Proof of Theorem 1

*Proof of Theorem 1.* By combining Lemma 1 and Proposition 1, we directly obtain following results. Under assumptions stated in Theorem 1, we have

$$\|g^{\pi_\star} - g^{\pi_K}\|_\infty \le C_\mu^{1/2} \frac{8\|Q^{\pi_\star}\|_{2,\mu}}{K + 2}$$
$$+ C_\mu^{1/2} \frac{2K}{3}\left(\sqrt{3\epsilon'} + \sqrt{\frac{60(K + 1)^2 R^2 \ln(2KN_{\epsilon'}^2/\delta)}{n}} + \max_{k=0,\dots,K-1} \sqrt{13\epsilon_B(\mathcal{F}_k, \mathcal{F}_{k+1})}\right),$$

$$\|g^{\pi_\star} - g^{\pi_K}\|_{2,\rho} \le C_{\mu,\rho}^{1/2} \frac{8\|Q^{\pi_\star}\|_{2,\mu}}{K+2}$$

$$+ C_{\mu,\rho}^{1/2} \frac{2K}{3} \left( \sqrt{3\epsilon'} + \sqrt{\frac{60(K+1)^2 R^2 \ln(2KN_{\epsilon'}^2/\delta)}{n}} + \max_{k=0,\dots,K-1} \sqrt{13\epsilon_B(\mathcal{F}_k, \mathcal{F}_{k+1})} \right),$$

where

$$N_\epsilon' = \max_{k=1,\dots,K} N_{k,\epsilon'}, \qquad N_{k,\epsilon} = \mathcal{N}\big(\tfrac{\epsilon'}{108(2k+1)R}; \mathcal{F}_k, \|\cdot\|_\infty \big), \quad \text{for } k=1,\dots,K.$$

Given $\epsilon > 0$, for the first inequality, let $K = \lceil 18 C_\mu^{1/2} \|Q^{\pi_\star}\|_{2,\mu}/\epsilon \rceil, \epsilon' = \frac{4\epsilon^2}{27K^2 C_\mu}, n = \frac{36K^2 C_\mu}{\epsilon^2} 60 R^2 (K+1)^2 \ln(2K\mathcal{N}_{\epsilon'}^2/\delta)$. Then, by direct calculation, we derive that

$$\|g^{\pi_\star} - g^{\pi_K}\|_\infty \le \epsilon + 3K C_\mu^{1/2} \max_{k=0,\dots,K-1} \sqrt{\epsilon_B(\mathcal{F}_k, \mathcal{F}_{k+1})}$$

with sample complexity

$$n = \mathcal{O}\left( \frac{\|Q^{\pi_\star}\|_{2,\mu}^4 C_\mu^3 R^2}{\epsilon^6} \ln(\mathcal{N}_\epsilon^2 C_\mu^{1/2}/(\delta\epsilon)) \right)$$

where

$$N_\epsilon = \max_{k=1,\dots,K} N_{k,\epsilon}, \qquad N_{k,\epsilon} = \mathcal{N}\big(\tfrac{\epsilon^4}{10^6 k C_\mu^2 \|Q^{\pi_\star}\|_{2,\mu}^2 R}; \mathcal{F}_k, \|\cdot\|_\infty \big), \quad \text{for } k=1,\dots,K.$$

Similarly, given $\epsilon > 0$, for second inequality, let $K = \lceil 18 C_{\mu,\rho}^{1/2} \|Q^{\pi_\star}\|_{2,\mu}/\epsilon \rceil, \epsilon' = \frac{4\epsilon^2}{27K^2 C_{\mu,\rho}}, n = \frac{36K^2 C_{\mu,\rho}}{\epsilon^2} 60 R^2 (K+1)^2 \ln(2K^2 \mathcal{N}_{\epsilon'}/\delta)$, and

$$\|g^{\pi_\star} - g^{\pi_K}\|_{2,\rho} \le \epsilon + 3K C_{\mu,\rho}^{1/2} \max_{k=0,\dots,K-1} \sqrt{\epsilon_B(\mathcal{F}_k, \mathcal{F}_{k+1})}$$

with sample complexity

$$n = \mathcal{O}\left( \frac{\|Q^{\pi_\star}\|_{2,\mu}^4 C_{\mu,\rho}^3 R^2}{\epsilon^6} \ln(\mathcal{N}_\epsilon^2 C_\mu^{1/2}/(\delta\epsilon)) \right)$$

where

$$N_\epsilon = \max_{k=1,\dots,K} N_{k,\epsilon}, \qquad N_{k,\epsilon} = \mathcal{N}\big(\tfrac{\epsilon^4}{10^6 k C_{\mu,\rho}^2 \|Q^{\pi_\star}\|_{2,\mu}^2 R}; \mathcal{F}_k, \|\cdot\|_\infty \big), \quad \text{for } k=1,\dots,K.$$

$\square$

### D.3 Proof of Lemma 2

We first introduce empirical covering number.

**Definition 5** (empirical covering number). *For a given function class $\mathcal{F}$ of real valued functions and set $x^{1:n} = (x_1, \dots, x_n)$, denote the covering number of $\mathcal{F}$ equipped with the empirical $l_1$ pseudo metric $l_{x^{1:n}}(f, g) = \frac{1}{n} \sum_{i=1}^n |f(x_i) - g(x_i)|$ by $\mathcal{N}(\epsilon, \mathcal{F}, x^{1:n})$.*

Although the empirical convering number depends on number of samples, but it can be bounded by pseudo dimension which depends on only function space and $\epsilon$ as following fact shows.

**Fact 3** ([35], Corollary 3). *For any $x^{1:n} = (x_1, \dots, x_n)$, any function class $\mathcal{F}$ of real-valued functions taking values in $[0, M]$ with pseudo-dimension $V_\mathcal{F} < \infty$, and any $\epsilon > 0$,*

$$\mathcal{N}(\epsilon, \mathcal{F}, l_{x^{1:N}}) \le e(V_\mathcal{F} + 1) \left( \frac{2eM}{\epsilon} \right)^{V_\mathcal{F}}.$$

Define $L(g, f) = \mathbb{E}_{s_i, a_i \sim \mu}[Var_{s_i' \sim P(\cdot|s_i, a_i)}(r(s_i, a_i) + \max f(s_i', a))] + \|g - Tf\|_{2,\mu}^2$ where $Var$ denotes variance with respect to $s_i'$, and $\hat{L}(g, f) = \frac{1}{n} \sum_{i=1}^n (g(s_i, a_i) - r(s_i, a_i) - \max_a f(s_i', a))^2$. Then, $\mathbb{E}[\hat{L}(g, f)] = L(g, f)$ and following lemma holds.

**Lemma 8.** $\|\hat{T}f - Tf\|_{2,\mu}^2 - \inf_{g\in\mathcal{G}}\|g - Tf\|_{2,\mu}^2 \le 2\sup_{g\in\mathcal{G}}|L(g,f) - \hat{L}(g,f)|.$

*Proof of Lemma 8.* $\|\hat{T}f - Tf\|_{2,\mu}^2 - \inf_{g\in\mathcal{G}}\|g - Tf\|_{2,\mu}^2 = L(\hat{T}f,f) - \inf_{g\in\mathcal{F}}L(g,f) = L(\hat{T}f,f) - \hat{L}(\hat{T}f,f) + \hat{L}(\hat{T}f,f) - \inf_{g\in\mathcal{G}}L(g,f) \le 2\sup_{f\in\mathcal{F}}|L(g,f) - \hat{L}(g,f)|$ by definition of $\hat{T}f$. $\qquad\square$

For $\{\hat{T}f_k, f_k\}_{k=0}^{K-1}$ of Anc-F-QI, previous lemma implies that

$$\|\hat{T}f_k - Tf_k\|_{2,\mu}^2 - \inf_{g\in\mathcal{G}}\|g - Tf_k\|_{2,\mu}^2 \le \sup_{f\in\mathcal{F}}(\|\hat{T}f - Tf\|_{2,\mu}^2 - \inf_{g\in\mathcal{G}}\|g - Tf\|_{2,\mu}^2)$$

$$\le 2\sup_{g\in\mathcal{G}, f\in\mathcal{F}}|L(g,f) - \hat{L}(g,f)|.$$

Define the function $l_{f,g} : \mathcal{S}\times\mathcal{A}\times[-R,R]\times\mathcal{S}\to\mathbb{R}$ as $l_{f,g}(s_i,a_i,r_i,s_{i+1}) = (f(s_i,a_i) - r_i - \max_a g(s_{i+1},a))^2$ and the function space $\mathcal{L}_{\mathcal{F},\mathcal{G}} = \{l_{f,g}\,|\,f\in\mathcal{F}, g\in\mathcal{G}\}$ and $\mathcal{G}_{max} = \{\max_a g(s,a)\,|\,g\in\mathcal{G}\}$. The pseudo dimension of $\mathcal{G}_{max}$ could be bounded by following Lemma.

**Lemma 9.** *Define* $\mathcal{G}_{max} = \{\max_{a\in\mathcal{A}}g(\cdot,a) : g\in\mathcal{G}\}$. $V_{\mathcal{G}_{max}} \le 2|\mathcal{A}|V_{\mathcal{G}}\log(3|\mathcal{A}|)$.

*Proof of Lemma 9.* By the definition of pseudo dimension, we have $V_{\mathcal{G}} \ge V_{\mathcal{G}^i}$ where $\mathcal{G}^i = \{g(x,a_i)\,|\,g\in\mathcal{G}\}$. Since $\max_{a\in\mathcal{A}}g(\cdot,a) \le 0 \iff \forall i\ g(\cdot,a_i) \le 0$, the claim follows from Lemma 3.2.3 of [10]. $\qquad\square$

Now, we are ready to prove Lemma 2.

*Proof of Lemma 2.* Let $\mathcal{F}\subset\{f:\mathcal{S}\times\mathcal{A}\to[-f_{max},f_{max}]\,|\,f\in B(S\times A)\}$ and $\mathcal{G}\subset\{g:\mathcal{S}\times\mathcal{A}\to[-g_{max},g_{max}]\,|\,g\in B(S\times A)\}$. Without loss of generality, $g_{max}\le f_{max}$.

By similar argument in proof of Proposition 4 of [16], $\{s_i,a_i,r_i\}$ is also $\beta$-mixing with the coefficient $\{\beta_i\}$ and this implies $\{s_i,a_i,r_i,s_{i+1}\}$ is also stationary $\beta$-mixing with coefficient $\{\beta_{i-1}\}$. By direct calculation, $|\hat{L}(f,g)| \le (2f_{max}+R)^2$. Now, we apply Fact 2 with $l(f,g)$ and $Z_i = (s_i,a_i,r_i,s_{i+1})$. Then, we get

$$P\left(\sup_{f\in\mathcal{F}, g\in\mathcal{G}}\left|\hat{L}(f,g) - L(f,g)\right| > \epsilon\right) \le 16\mathbb{E}[\mathcal{N}(\epsilon/8, \mathcal{L}_{\mathcal{F},\mathcal{G}}, (Z_t')_{t\in H})]e^{-\frac{m_N\epsilon^2}{128(2f_{max}+R)^4}} + 2m_N\beta_{k_N}.$$

Since

$$\hat{L}(f_1,g_1) - \hat{L}(f_2,g_2)$$

$$= \frac{1}{n}\left|\sum_{i=1}^n (f_1(s_i,a_i) - r(s_i,a_i) - \max_{a\in\mathcal{A}}g_1(s_{i+1},a))^2 - \sum_{i=1}^n (f_2(s_i,a_i) - r(s_i,a_i) - \max_{a\in\mathcal{A}}g_2(s_{i+1},a))^2\right|$$

$$\le 2\frac{2f_{max}+R}{n}\sum_{i=1}^n(|f_1(s_i,a_i) - f_2(s_i,a_i)| + |\max_{a\in\mathcal{A}}g_1(s_{i+1},a) - \max_{a\in\mathcal{A}}g_2(s_{i+1},a)|),$$

this implies that

$$\mathcal{N}(4(2f_{max}+R)\epsilon, \mathcal{L}_{\mathcal{F},\mathcal{G}}, (z^{1:n}) \le \mathcal{N}(\epsilon,\mathcal{F}, s^{2:n+1})\mathcal{N}(\epsilon, \mathcal{G}_{max}, (s,a)^{1:n})$$

where $z_i = (s_i,a_i,r_i,s_{i+1})$ by definition of empirical covering number. Finally, by Fact 3, we get

$$\mathcal{N}(\epsilon/8, \mathcal{L}_{\mathcal{F},\mathcal{G}}, (Z_t')_{t\in H})$$

$$\le e(V_{\mathcal{F}}+1)\left(\frac{128(2f_{max}+R)e}{\epsilon}\right)^{V_{\mathcal{F}}} e(V_{\mathcal{F}_{max}}+1)\left(\frac{128(2f_{max}+R)e}{\epsilon}\right)^{V_{\mathcal{G}_{max}}}$$

$$= C\left(\frac{1}{\epsilon}\right)^{V_{\mathcal{F}}+V_{\mathcal{G}_{max}}}$$

where $C = e^2(V_{\mathcal{F}}+1)(V_{\mathcal{G}_{max}}+1)(128(2f_{max}+R)e)^{V_{\mathcal{F}}+V_{\mathcal{G}_{max}}}$.

For calculation, we use following prior result.

**Fact 4** ([4], Lemma 14). *Let* $\beta_m \leq \bar{\beta}e^{(-bm^\kappa)}, N \geq 1, k_N = \lceil (C_2 N\epsilon^2/b)^{\frac{1}{1+\kappa}} \rceil, m_N = N/(2k_N), 0 < \delta \leq 1, V \geq 2$ *and* $C_1, C_2, \bar{\beta}, b, \kappa > 0$. *Define* $\epsilon$ *and* $C_0$ *as*

$$\epsilon = \sqrt{\frac{C_0(\max\{C_0/b, 1\})^{1/\kappa}}{C_2 N}}$$

*with* $C_0 = V/2 \log N + \log(e/\delta) + \log(\max(C_1 C_2^{V/2}, \bar{\beta}, 1))$

$$C_1 \left(\frac{1}{\epsilon}\right)^V e^{-4C_2 m_N \epsilon^2} + 2m_N \beta_{k_N} \leq \delta.$$

Then, by this fact and previous arguments, for $\epsilon = \sqrt{\frac{c_0(\max\{c_0/b,1\})^{1/\kappa}}{c_2 n}}$,

$$P\left(\sup_{f \in \mathcal{F}, g \in \mathcal{G}} \left|\hat{L}(f, g) - L(f, g)\right| \leq \epsilon\right) \geq 1 - \delta$$

where $c_0 = (V_\mathcal{F} + V_{\mathcal{G}_{max}})/2 \log n + \log(e/\delta) + \log(\max(c_1 c_2^{(V_\mathcal{F}+V_{\mathcal{G}_{max}})/2}, \bar{\beta}, 1), c_1 = 16e^2(V_\mathcal{F} + 1)(V_{\mathcal{G}_{max}} + 1)(128(2f_{max} + R)e2)^{V_\mathcal{F}+V_{\mathcal{G}_{max}}}, c_2 = \frac{1}{512(2f_{max}+R)^4}, V_{\mathcal{G}_{max}} = 2|\mathcal{A}|V_\mathcal{G} \log(3|\mathcal{A}|)$. Let $\mathcal{G} = \mathcal{F}_k, \mathcal{F} = \mathcal{F}_{k+1}$ and $g = f_k$. By Lemma 8, this implies that with $1 - \delta$ probability,

$$\|Tf_k - \hat{T}f_k\|_{\mu,2}^2 \leq \epsilon_B + \sqrt{\frac{c_0(\max\{c_0/b,1\})^{1/\kappa}}{4c_2 n}}.$$

Finally, by manipulating $\delta$, we get desired result. $\qquad\square$

### D.4 Proof of Theorem 2

*Proof of Theorem 2.* By combining Lemma 2 and Proposition 1, we directly obtain following results. Under assumptions stated in Theorem 2, we have

$$\|g^{\pi_\star} - g^{\pi_K}\|_\infty \leq C_\mu^{1/2} \frac{8\|Q^{\pi_\star}\|_{2,\mu}}{K+2}$$
$$+ C_\mu^{1/2} \frac{2K}{3} \left( \left(\frac{c_{0,K}(\max\{c_{0,K}/b, 1\})^{1/\kappa}}{c_{2,K} n}\right)^{1/4} + \max_{k=0,\dots,K-1} \sqrt{\epsilon_B(\mathcal{F}_k, \mathcal{F}_{k+1})} \right),$$

$$\|g^{\pi_\star} - g^{\pi_K}\|_{2,\rho} \leq C_{\mu,\rho}^{1/2} \frac{8\|Q^{\pi_\star}\|_{2,\mu}}{K+2}$$
$$+ C_{\mu,\rho}^{1/2} \frac{2K}{3} \left( \left(\frac{c_{0,K}(\max\{c_{0,K}/b, 1\})^{1/\kappa}}{c_{2,K} n}\right)^{1/4} + \max_{k=0,\dots,K-1} \sqrt{\epsilon_B(\mathcal{F}_k, \mathcal{F}_{k+1})} \right),$$

where $c_{0,K} = \max_{k=0,\dots,K-1} c_{0,k}, c_{0,k} = (V_{\mathcal{F}_{k+1}} + V_{(\mathcal{F}_k)_{max}})/2 \log n + \log(e/(K\delta)) + \log(\max(c_{1,k}, \bar{\beta}, 1)), c_{1,k} = 16e^2(V_{\mathcal{F}_{k+1}} + 1)(V_{(\mathcal{F}_k)_{max}} + 1)(24e)^{V_{\mathcal{F}_{k+1}}+V_{(\mathcal{F}_k)_{max}}}, c_{2,K} = \frac{1}{512(2K+1)^4 R^4}, V_{(\mathcal{F}_k)_{max}} = 2|\mathcal{A}|, V_{\mathcal{F}_k} \log(3|\mathcal{A}|)$.

Given $\epsilon > 0$, for the first inequality, let $K = \lceil 9C_\mu^{1/2} \|Q^{\pi_\star}\|_{2,\mu}/\epsilon \rceil$. Then, by direct calculation, we derive that
$$\|g^{\pi_\star} - g^{\pi_K}\|_\infty \leq \epsilon + KC_\mu^{1/2} \max_{k=0,\dots,K-1} \sqrt{\epsilon_B(\mathcal{F}_k, \mathcal{F}_{k+1})}$$

with sample complexity

$$n = \tilde{\mathcal{O}}\left(\frac{b^{-1/\kappa}(c'_{0,K})^{\frac{1+\kappa}{\kappa}} R^4 \|Q^{\pi_\star}\|_{2,\mu}^8 C_\mu^6}{\epsilon^{12}}\right)$$

where $c'_{0,K} = \max_{k=0,\dots,K-1} c'_{0,k}, c'_{0,k} = \log(1/\delta) + \log(\max(c_{1,k}, \bar{\beta})), c_{1,k} = 16e^2(V_{\mathcal{F}_{k+1}} + 1)(V_{(\mathcal{F}_k)_{max}} + 1)(24e)^{V_{\mathcal{F}_{k+1}}+V_{(\mathcal{F}_k)_{max}}}, V_{(\mathcal{F}_k)_{max}} = 2|\mathcal{A}|, V_{\mathcal{F}_k} \log(3|\mathcal{A}|)$, and $\tilde{\mathcal{O}}$ ignores all logarithmic factors.

Similarly, given $\epsilon > 0$, for the second inequality, let $K = \lceil 9C_{\mu,\rho}^{1/2} \|Q^{\pi_\star}\|_{2,\mu}/\epsilon \rceil$. Then, by direct calculation, we derive that

$$\|g^{\pi_\star} - g^{\pi_K}\|_\infty \le \epsilon + KC_{\mu,\rho}^{1/2} \max_{k=0,\ldots,K-1} \sqrt{\epsilon_B(\mathcal{F}_k, \mathcal{F}_{k+1})}$$

with sample complexity

$$n = \tilde{\mathcal{O}} \left( \frac{b^{-1/\kappa}(c'_{0,K})^{\frac{1+\kappa}{\kappa}} R^4 \|Q^{\pi_\star}\|_{2,\mu}^8 C_{\mu,\rho}^6}{\epsilon^{12}} \right)$$

where $c'_{0,K} = \max_{k=0,\ldots,K-1} c'_{0,k}, c'_{0,k} = \log(1/\delta) + \log(\max(c_{1,k}, \bar{\beta})), c_{1,k} = 16e^2(V_{\mathcal{F}_{k+1}} + 1)(V_{(\mathcal{F}_k)_{max}} + 1)(24e)^{V_{\mathcal{F}_{k+1}} + V_{(\mathcal{F}_k)_{max}}}, V_{(\mathcal{F}_k)_{max}} = 2|\mathcal{A}|, V_{\mathcal{F}_k} \log(3|\mathcal{A}|)$, and $\tilde{\mathcal{O}}$ ignores all logarithmic factors. $\qquad\square$

# E  Omitted proofs in Section 5

## E.1  Proof of Theorem 3

We first prove following key lemma.

**Lemma 10.** *Assume Assumptions 1, 2, 3, 8, 11, and 12 (Bellman optimality equation, existence of argmin, star-shaped function space, normalized function space, range of function space, IID dataset). Let $\mu$ be the distribution generating the dataset. Let $\epsilon > 0$ and $\delta > 0$. With probability $1 - \delta$, $\{f_k, \hat{T} f_k\}_{k=0}^{K-1}$ of R-Anc-F-QI satisfies*

$$\|Tf_k - \hat{T} f_k\|_{\mu,2}^2 \le \frac{30(R + 4\|Q^{\pi_\star}\|_\infty)^2 \ln(2KN_\epsilon^2/\delta)}{n} + 3\epsilon + 13\epsilon_B(\mathcal{F}, \mathcal{F}),$$

*where*

$$N_\epsilon = \mathcal{N}(\tfrac{\epsilon}{108(R+4\|Q^{\pi_\star}\|_\infty)}; \mathcal{F}, \|\cdot\|_\infty).$$

*Proof.* The proof basically follows from the proof of Lemma 1. $\qquad\square$

Now, we prove Theorem 3.

*Proof of Theorem 3.* Consider Apporximate Relative Anchored Value Iteration

$$Q_r^k = (1 - \lambda_k)Q_r^0 + \lambda_k(TQ_r^{k-1} + \epsilon_k - c_k \mathbf{1}) \qquad \text{(Apx-R-Anc-QI)}$$

for $c_k \in \mathbb{R}$. Also, consider corresponding Approximate Anchored Value Iteration with same $\epsilon_k$ and starting point $Q_r^0$

$$Q^k = (1 - \lambda_k)Q_r^0 + \lambda_k(TQ^{k-1} + \epsilon_k). \qquad \text{(Apx-Anc-QI)}$$

Since $Q^k - Q_r^k = d_k \mathbf{1}$ for some $d_k \in \mathbb{R}$, $\max_a Q^k(s,a) = \max_a Q_r^k(s,a)$ for all $s \in \mathcal{S}$ by the defintion of Bellman operator and this implies induced policies are same. Thus, Proposition 1 also holds for Apx-R-Anc-QI.

By combining Lemma 10 and Proposition 1, we directly obtain following results. Under assumptions stated in Theorem 3,

$$\|g^{\pi_\star} - g^{\pi_K}\|_\infty \le C_\mu^{1/2} \frac{8\|Q^{\pi_\star}\|_{2,\mu}}{K+2}$$
$$+ C_\mu^{1/2} \frac{2K}{3} \left( \sqrt{3\epsilon'} + \sqrt{\frac{30(R + 4\|Q^{\pi_\star}\|_\infty)^2 \ln(2KN_{\epsilon'}^2/\delta)}{n}} + \sqrt{13\epsilon_B(\mathcal{F}, \mathcal{F})} \right).$$

$$\|g^{\pi_\star} - g^{\pi_K}\|_{2,\rho} \le C_{\mu,\rho}^{1/2} \frac{8\|Q^{\pi_\star}\|_{2,\mu}}{K+2}$$
$$+ C_{\mu,\rho}^{1/2} \frac{2K}{3} \left( \sqrt{3\epsilon'} + \sqrt{\frac{30(R + 4\|Q^{\pi_\star}\|_\infty)^2 \ln(2KN_{\epsilon'}^2/\delta)}{n}} + \sqrt{13\epsilon_B(\mathcal{F}, \mathcal{F})} \right),$$

where

$$N_{\epsilon'} = \mathcal{N}\big(\tfrac{\epsilon'}{108(R+4\|Q^{\pi^\star}\|_\infty)}; \mathcal{F}, \|\cdot\|_\infty\big).$$

Given $\epsilon > 0$, for the first inequality, let $K = \lceil 18C_\mu^{1/2}\|Q^{\pi^\star}\|_{2,\mu}/\epsilon \rceil, \epsilon' = \tfrac{4\epsilon^2}{27K^2C_\mu}, n = \tfrac{36K^2C_\mu}{\epsilon^2}30(R+4\|Q^{\pi^\star}\|_\infty)^2\ln(2KN_{\epsilon'}^2/\delta)$. Then, by direct calculation, we derive that

$$\|g^{\pi^\star} - g^{\pi_K}\|_\infty \le \epsilon + 3KC_\mu^{1/2}\sqrt{\epsilon_B(\mathcal{F},\mathcal{F})}$$

with sample complexity

$$n = \mathcal{O}\left(\frac{(R+\|Q^{\pi^\star}\|_\infty)^2\,\|Q^{\pi^\star}\|_\infty^2\,C_\mu^3}{\epsilon^4}\ln(\mathcal{N}_\epsilon^2 C_\mu^{1/2}/(\delta\epsilon))\right)$$

where

$$N_\epsilon = \mathcal{N}\big(\tfrac{\epsilon^4}{10^6 C_\mu^2 (R+\|Q^{\pi^\star}\|_\infty)\|Q^{\pi^\star}\|_\infty^2}; \mathcal{F}, \|\cdot\|_\infty\big).$$

Similarly, given $\epsilon > 0$, for second inequality, let $K = \lceil 18C_{\mu,\rho}^{1/2}\|Q^{\pi^\star}\|_{2,\mu}/\epsilon \rceil, \epsilon' = \tfrac{4\epsilon^2}{27K^2C_{\mu,\rho}}, n = \tfrac{36K^2C_{\mu,\rho}}{\epsilon^2}30(R+4\|Q^{\pi^\star}\|_\infty)^2\ln(2K^2\mathcal{N}_{\epsilon'}/\delta)$, and

$$\|g^{\pi^\star} - g^{\pi_K}\|_{2,\rho} \le \epsilon + 3KC_{\mu,\rho}^{1/2}\sqrt{\epsilon_B(\mathcal{F},\mathcal{F})}$$

with sample complexity

$$n = \mathcal{O}\left(\frac{(R+\|Q^{\pi^\star}\|_\infty)^2\,\|Q^{\pi^\star}\|_\infty^2\,C_{\mu,\rho}^3}{\epsilon^4}\ln(\mathcal{N}_\epsilon^2 C_{\mu,\rho}^{1/2}/(\delta\epsilon))\right)$$

where

$$N_\epsilon = \mathcal{N}\big(\tfrac{\epsilon^4}{10^6 C_{\mu,\rho}^2 (R+\|Q^{\pi^\star}\|_\infty)\|Q^{\pi^\star}\|_\infty^2}; \mathcal{F}, \|\cdot\|_\infty\big).$$

$\square$

### E.2 Proof of Theorem 4

We first prove following key Lemma.

**Lemma 11.** *Assume Assumptions 1, 2, 3, 9, 10, 11, and 12 (Bellman optimality equation, existence of argmin, star-shaped function space, normalized function space, range of function space, single-trajectory dataset, $\beta$-mixing single-trajectory). Let $\mu$ be the distribution generating the dataset defined as $\mu(s,a) = \nu(s)\pi_b(a\,|\,s)$. Let $\epsilon > 0$ and $\delta > 0$. With probability $1 - \delta$, $\{f_k, \hat{T}f_k\}_{k=0}^{K-1}$ of R-Anc-F-QI satisfies*

$$\|Tf_k - \hat{T}f_k\|_{\mu,2}^2 \le \epsilon_B(\mathcal{F},\mathcal{F}) + \sqrt{\frac{c_0(\max\{c_0/b,1\})^{1/\kappa}}{c_2 n}}$$

*where $c_0 = (V_\mathcal{F} + V_{\mathcal{F}_{max}})\log n/2 + \log(e/(K\delta)) + \log(\max(c_1,\bar\beta)), c_1 = 16e^2(V_\mathcal{F}+1)(V_{\mathcal{F}_{max}}+1)(24e)^{V_\mathcal{F}+V_{\mathcal{F}_{max}}}, c_2 = \tfrac{1}{512(R+4\|Q^{\pi^\star}\|_\infty)^4}, V_{\mathcal{F}_{max}} = 2|\mathcal{A}|V_\mathcal{F}\log(3|\mathcal{A}|)$.*

*Proof.* The proof basically follows from the proof of Lemma 2. $\square$

Now, we prove Theorem 4.

*Proof of Theorem 4.* By combining Lemma 11 and Proposition 1, we directly obtain following results. Under assumptions stated in Theorem 11, we have

$$\|g^{\pi^\star} - g^{\pi_K}\|_\infty \le C_\mu^{1/2}\frac{8\|Q^{\pi^\star}\|_{2,\mu}}{K+2}$$
$$+ C_\mu^{1/2}\frac{2K}{3}\left(\left(\frac{c_0(\max\{c_0/b,1\})^{1/\kappa}}{c_2 n}\right)^{1/4} + \sqrt{\epsilon_B(\mathcal{F},\mathcal{F})}\right),$$

$$\|g^{\pi_\star} - g^{\pi_K}\|_{2,\rho} \leq C_{\mu,\rho}^{1/2} \frac{8\|Q^{\pi_\star}\|_{2,\mu}}{K+2}$$
$$+ C_{\mu,\rho}^{1/2} \frac{2K}{3} \left( \left( \frac{c_0(\max\{c_0/b,1\})^{1/\kappa}}{c_2 n} \right)^{1/4} + \sqrt{\epsilon_B(\mathcal{F}, \mathcal{F})} \right),$$

where $c_0 = (V_{\mathcal{F}} + V_{\mathcal{F}_{max}})/2 \log n + \log(e/(K\delta)) + \log(\max(c_1, \bar{\beta}, 1))$, $c_1 = 16e^2(V_{\mathcal{F}}+1)(V_{\mathcal{F}_{max}}+1)(24e)^{V_{\mathcal{F}}+V_{\mathcal{F}_{max}}}$, $c_2 = \frac{1}{512(R+4\|Q^{\pi_\star}\|_\infty)^4}$, $V_{\mathcal{F}_{max}} = 2|\mathcal{A}|V_{\mathcal{F}}\log(3|\mathcal{A}|)$.

Given $\epsilon > 0$, for the first inequality, let $K = \lceil 9C_\mu^{1/2}\|Q^{\pi_\star}\|_{2,\mu}/\epsilon \rceil$. Then, by direct calculation, we derive that
$$\|g^{\pi_\star} - g^{\pi_K}\|_\infty \leq \epsilon + KC_\mu^{1/2}\sqrt{\epsilon_B(\mathcal{F}, \mathcal{F})}$$
with sample complexity

$$n = \tilde{\mathcal{O}} \left( \frac{b^{-1/\kappa}(c_0')^{\frac{1+\kappa}{\kappa}}(R+\|Q^{\pi_\star}\|_\infty)^4\|Q^{\pi_\star}\|_\infty^4 C_\mu^4}{\epsilon^8} \right)$$

where $c_0' = \log(1/\delta) + \log(\max(c_1, \bar{\beta}))$, $c_1 = 16e^2(V_{\mathcal{F}}+1)(V_{\mathcal{F}_{max}}+1)(24e)^{V_{\mathcal{F}}+V_{\mathcal{F}_{max}}}$, $V_{\mathcal{F}_{max}} = 2|\mathcal{A}|V_{\mathcal{F}}\log(3|\mathcal{A}|)$, and $\tilde{\mathcal{O}}$ ignores all logarithmic factors.

Similarly, given $\epsilon > 0$, for the second inequality, let $K = \lceil 9C_{\mu,\rho}^{1/2}\|Q^{\pi_\star}\|_{2,\mu}/\epsilon \rceil$. Then, by direct calculation, we derive that
$$\|g^{\pi_\star} - g^{\pi_K}\|_\infty \leq \epsilon + KC_{\mu,\rho}^{1/2}\sqrt{\epsilon_B(\mathcal{F}, \mathcal{F})}$$
with sample complexity

$$n = \tilde{\mathcal{O}} \left( \frac{b^{-1/\kappa}(c_0')^{\frac{1+\kappa}{\kappa}}(R+\|Q^{\pi_\star}\|_\infty)^4\|Q^{\pi_\star}\|_\infty^4 C_{\mu,\rho}^4}{\epsilon^8} \right)$$

where $c_0' = \log(1/\delta) + \log(\max(c_1, \bar{\beta}))$, $c_1 = 16e^2(V_{\mathcal{F}}+1)(V_{\mathcal{F}_{max}}+1)(24e)^{V_{\mathcal{F}}+V_{\mathcal{F}_{max}}}$, $V_{\mathcal{F}_{max}} = 2|\mathcal{A}|V_{\mathcal{F}}\log(3|\mathcal{A}|)$, and $\tilde{\mathcal{O}}$ ignores all logarithmic factors. $\square$

