# OpenReview forum: "Finite-Time Bounds for Average-Reward Fitted Q-Iteration"
_NeurIPS.cc/2025/Conference — NeurIPS 2025 poster_

### Official Review · Reviewer_s4WQ · 2025-06-11

**Clarity:** 2
**Significance:** 3
**Originality:** 3
**Rating:** 4
**Confidence:** 4

**Summary:**

This paper presents the first finite-sample complexity results for offline reinforcement learning (RL) in the average-reward setting with general function approximation. The authors propose a novel algorithm, Anchored Fitted Q-Iteration (Anc-F-QI), which introduces an anchoring mechanism-akin to weight decay-to ensure finite-time convergence.

They derive sample complexity bounds for two scenarios:
1. IID data: $\tilde{O}(1/\epsilon^6)$
2. Single-trajectory data (with $\beta$-mixing): $\tilde{O}(1/\epsilon^{10})$


To improve these bounds, the authors propose a normalized variant, Relative Anchored Fitted Q-Iteration (R-Anc-F-QI), which further improves the bounds to:
1.  IID: $\tilde{O}(1/\epsilon^4)$
2.  Single-trajectory: $\tilde{O}(1/\epsilon^8)$


All results are derived under the mild and general assumption that the underlying MDP is weakly communicating, extending beyond the ergodic or unichain assumptions used in prior work.

**Questions:**

1. Why is the weakly communicating assumption less restrictive than ergodic or unichain assumptions? What practical implications does this have?
  2.  What role does the anchoring mechanism play in ensuring finite-time convergence in the average-reward setting?
  3.  How does the normalization in R-Anc-F-QI (relative version) improve sample complexity compared to standard Anc-F-QI?
  4.  How does the sample complexity change under IID vs.  single-trajectory data? What causes the higher complexity in the latter case?
  5.  How critical is the full-coverage assumption for the derived bounds? Could the results extend to partial-coverage settings without sacrificing generality?
  6.  Are there known practical scenarios or benchmarks where average-reward RL is preferable to discounted-return RL, justifying the relevance of this work?

**Ethical Concerns:**

["NO or VERY MINOR ethics concerns only"]

**Limitations:**

Yes

**Paper Formatting Concerns:**

No concerns

**Quality:**

3

**Strengths And Weaknesses:**

Strengths
  1. Provides the first finite-time sample complexity guarantees for average-reward offline RL under weakly communicating MDPs.
  2. Works under more general conditions than prior literature (e.g., ergodic or unichain MDPs).
  3.  Introduces an anchoring mechanism within fitted Q-iteration and a normalization technique that improves convergence.
  4.  Achieves better bounds with the normalized variant (R-Anc-F-QI).
  5.  Clearly states all assumptions and provides complete theoretical proofs.
  6.  Includes a concise comparison with prior work, highlighting improvements.


Weaknesses
  1. The paper lacks experiments, limiting insights into the practical performance of the proposed methods.
  2.  Assumes full state-action coverage in offline datasets, which is difficult to guarantee in practice.
  3.  The paper is mathematically dense and may be difficult for non-experts to follow.
  4.   Realistic implications of assumptions (e.g., $\beta$-mixing, Bellman completeness) are not deeply explored.
  5.  Requires Bellman completeness to ensure zero inherent Bellman error, which may not hold with general function approximators.

---

> ### Author Rebuttal · Authors · 2025-07-31
>
> We thank the reviewer for detailed feedback.
>
> Questions
>
> (i) Intuitively speaking, a weakly communicating MDP requires that one policy be good, whereas a unichain MDP requires every policy to be good. By definition [1], an MDP is weakly communicating if there exists a policy for which set of states where every state in the set is accessible from every other state (plus a possibly empty set of transient states), and an MDP is unichain if, for all policies, the induced Markov chain has a single recurrent class (plus a possibly empty set of transient states). As we clarified in the introduction, this difference induces that a weakly communicating MDP guarantees the Bellman equation only for the optimal policy, while a unichain MDP satisfies the Bellman equation for every policy. In this sense, weakly communicating assumption is less restrictive than ergodic or unichain assumption, and many practical algorithms have been developed for various settings of weakly communicating MDPs [2,3].
>
>
> (ii) Roughly speaking, in the anchoring mechanism, the anchor term pulls the iterates toward the starting point, and this gives the effect of acceleration in convergence. In the fixed‑point literature, it is known that when the anchoring mechanism is combined with fixed‑point iteration, it effectively reduces the residual error. Recently, [4,5] have shown that anchoring indeed provides accelerated rates and efficient sample complexity in the average reward setup. In our analysis, the anchoring mechanism ensures sublinear convergence with respect to both the Bellman error and the policy as shown in the proof of Proposition 1, and this lead to first finite time bounds for  weakly communicating MDPs.
>
> (iii) The role of relative normalization in R‑Anc‑F‑QI is to bound the norm of the function approximator, $\|f_k\|$, from diverging, as our function space range assumption states. Compared to Anc-F-QI, this normalized technique leads to efficient sample complexity. This is because, in the theoretical analysis, we used Bernstein’s and Pollard’s inequalities, whose upper bounds are proportional to the square of $\|f_k\|$. Thus, R‑Anc‑F‑QI improves sample complexity, as we showed in Theorems 3 and 4.
>
>
> (iv) As the reviewer pointed out, our theorems show that the sample complexity of an IID dataset is more efficient than that of a single‑trajectory dataset. We believe that this discrepancy arises because the inequalities we applied differ in the analysis. As shown in the proofs of Lemmas 5 and 11, we used Bernstein’s inequality for the IID dataset and Pollard’s tail inequality generalized to mixing data for the single‑trajectory dataset, and Pollard’s tail inequality yields looser bounds than Bernstein’s inequality. Consequently, the sample complexity for a mixing single‑trajectory dataset is less efficient than for an IID dataset. We briefly note that this gap also exists in the discounted‑reward offline RL setting [7,8].
>
>
> (v) Our analysis of Anc-F-QI uses the full coverage condition to bound the products of transition matrices induced by policies. As shown in the proof of Proposition 1, the policy error of Approximate Anchored Q‑Iteration is bounded by a sum of such products with products of coefficients due to the anchoring mechanism. To control these terms, the full coverage is needed and this is how standard Fitted Q‑Iteration analyses in the discounted reward setting use the full coverage condition [7,9].
>
>  We expect that our anchoring mechanism can provide finite sample complexity guarantees under partial coverage condition by incorporating a pessimistic approach. A series of works [10–12] has shown that penalizing the estimated value or policy according to its uncertainty relaxes the need for full coverage condition. In particular, [12] introduced Pessimistic Value Iteration, which subtracts an uncertainty term scaled by a confidence parameter from the estimated Q‑value, and achieve finite bounds under partial coverage condition. Referring to those prior works, by combining our anchoring mechanism with this pessimistic approach,  we believe that our analysis can be extended to the partial coverage setting.
>
> (vi) For continual or infinite‑horizon tasks where long‑term performance is significant, the average reward setup is a preferable alternative to the discounted reward setup [13-15]. Specifically, average reward framework suits problems like robotic locomotion [16] and queuing systems [17], and many practical RL algorithms have been studied for this setup [18,19].
>
> References
>
> [1] M. L. Puterman. Markov Decision Processes: Discrete Stochastic Dynamic Programming. John Wiley and Sons, 2014
>
> [2] Bartlett, Peter L., and Ambuj Tewari. REGAL: A regularization based algorithm for reinforcement learning in weakly communicating MDPs. arXiv preprint arXiv:1205.2661 (2012).
>
> [3] Wan, Yi, and Richard S. Sutton. On convergence of average-reward off-policy control algorithms in weakly communicating MDPs. arXiv preprint arXiv:2209.15141 (2022).
>
> [4] J. Lee and E. Ryu. Optimal non-asymptotic rates of value iteration for average-reward MDPs International Conference on Learning Representations, 2025.
>
> [5] J. Lee, M. Bravo, and R. Cominetti. Near-optimal sample complexity for MDPs via anchoring. Interantional Conference on Machine Learning, 2025.
>
> [6] D. J. White. Dynamic programming, Markov chains, and the method of successive approximations. J. Math. Anal. Appl, 6(3):373–376, 1963.
>
> [7] J. Chen and N. Jiang. Information-theoretic considerations in batch reinforcement learning. International Conference on Machine Learning, 2019.
>
> [8] A. Antos, C. Szepesvári, and R. Munos. Learning near-optimal policies with bellman-residual minimization based fitted policy iteration and a single sample path. Machine Learning, 2008.
>
> [9] R. Munos and C. Szepesvári. Finite-time bounds for fitted value iteration. Journal of Machine Learning Research, 2008
>
> [10] Yao Liu, Adith Swaminathan, Alekh Agarwal, and Emma Brunskill. Provably good batch off-policy reinforcement learning without great exploration. Neural Information Processing Systems, 2020
>
> [11] T. Xie, C.-A. Cheng, N. Jiang, P. Mineiro, and A. Agarwal. Bellman-consistent pessimism for offline reinforcement learning. Neural Information Processing Systems, 2021
>
> [12]  Y. Jin, Z. Yang, and Z. Wang. Is pessimism provably efficient for offline RL? International Conference on Machine Learning, 2021
>
> [13] Sutton, R. S., Barto, A. G. (1998). Reinforcement learning: An introduction. Cambridge: MIT press.
>
> [14] Mahadevan, S. (1996). Average reward reinforcement learning: Foundations, algorithms, and empirical results. Machine learning, 22(1), 159-195.
>
> [15] Naik, A., Shariff, R., Yasui, N., Yao, H., Sutton, R. S. (2019). Discounted reinforcement learning is not an optimization problem. arXiv preprint arXiv:1910.02140.
>
> [16] Kober, J., Bagnell, J. A.,  Peters, J. (2013). Reinforcement learning in robotics: A survey. The International Journal of Robotics Research, 32(11), 1238-1274.
>
> [17] Schneckenreither, Manuel, Stefan Haeussler, and Juanjo Peiro. Average reward adjusted deep reinforcement learning for order release planning in manufacturing. Knowledge-based systems 247 (2022): 108765.
>
>
> [18] Hisaki, Yukinari, and Isao Ono. RVI-SAC: average reward off-policy deep reinforcement learning. arXiv preprint arXiv:2408.01972. 2024
>
> [19] Zhang, Yiming, and Keith W. Ross. On-policy deep reinforcement learning for the average-reward criterion. International Conference on Machine Learning. 2021.

---

> > ### Comment · Reviewer_s4WQ · 2025-08-05
> >
> > I appreciate the authors' response during the rebuttal period and will maintain my positive assessment, as the paper’s strengths continue to outweigh its weaknesses.

---

### Official Review · Reviewer_vmhP · 2025-07-01

**Clarity:** 2
**Significance:** 2
**Originality:** 2
**Rating:** 4
**Confidence:** 3

**Summary:**

This paper analyzes the average-reward RL setting for MDPs with finite state and action spaces. For this setting, the authors propose a variant of the fitted-Q iteration algorithm that uses an anchor to stabilize the iterates generated by the algorithm. Specifically, the anchor amounts to taking a convex combination of an initial iterate and the new estimate of the Bellman optimality operator; the weights of the convex combination are appropriately tuned to achieve finite-time rates. The authors establish sample-complexity bounds under both i.i.d. and single-trajectory, Markovian sampling (with standard mixing assumptions).

**Questions:**

My questions and comments are listed under "strengths and weaknesses".

**Ethical Concerns:**

["NO or VERY MINOR ethics concerns only"]

**Final Justification:**

The authors provided a detailed rebuttal that addressed all my initial concerns. As such, I am happy to raise my score.

**Limitations:**

Yes.

**Paper Formatting Concerns:**

There are no formatting concerns.

**Quality:**

2

**Strengths And Weaknesses:**

**Strengths**

- While there has been a lot of recent work analyzing the infinite-horizon discounted RL setting, analogous results for the average reward case are rarer in the literature. This work takes a step towards closing this gap by considering general function approximators and working under weaker assumptions on the underlying MDP.

- It seems that this is the first paper to analyze the anchored variant of fitted Q-iteration with function approximation.

- Finite-time results are established for the Markov noise setting, which is more challenging to analyze relative to i.i.d. sampling.

**Weaknesses**

I have several comments and questions regarding the nature of the contributions and the assumptions made in the paper. These are outlined below.

- If I understand correctly, based on the discussion in Section 2.1., the idea of anchoring does not seem to be unique to this paper. The idea seems to have been already explored for tabular settings. Is the main algorithmic contribution to extend it to account for function approximators? I would like to have more clarity in terms of what is novel in terms of algorithmic development.

- The assumption that the range of the function classes needs to expand with time (Assumption 7) seems quite strong and non-standard  to me. Can the authors point to any other papers that have made similar assumptions?

The reason behind making this assumption seems to stem from a crude bound on $ \Vert f_k \Vert $ that is on the order of $O(k)$, where $k$ is the iteration count. Can a finer analysis be done to show that $ \Vert f_k \Vert $  is perhaps $O(1)$? I presume the sequence ${f_k}$ will eventually converge close to the "best fit" function, and as such, its magnitude should not keep increasing linearly with time.

-  The final result in Theorem 1 seems to suggest convergence to a ball of size $O(\epsilon) + O(K)$ around the optimal state-action value function. My concern is about the second term of order $O(K)$, (where $K$ is the number of iterations) that might make the overall bound vacuous. Since the range of the function classes is required to expand with time, can the authors comment on the magnitude of the term hitting the $O(K)$ term?

- There is no discussion at all regarding the tightness of the sample-complexity bounds derived in this paper. Sample-complexity bounds on the order of $\tilde{O}(1/\epsilon^6)$ and $\tilde{O}(1/\epsilon^{10})$ under i.i.d. and Markov noise, respectively, seem quite pessimistic (at least compared to what is known in the discounted setting). I would like to see (i) a clear discussion of how these bounds compare with prior results for the average reward setting (even if they are derived under stronger assumptions), and (ii) some discussion of lower bounds/optimality.

- Why is there a substantial gap in the sample-complexity bounds derived for the i.i.d. and Markov settings? For infinite-horizon discounted problems, this gap is only a logarithmic factor that captures the mixing time of the underlying Markov chain; see Bhandari et al., COLT 2018 for instance. In this paper, the gap seems to be on the order of $\tilde{O}(1/\epsilon^4)$. Is this just an artifact of the analysis or is there a deeper reason?

- While I appreciate the fact that the authors derived results for the Markov noise setting, similar results do exist for the discounted setting.  For geometrically mixing Markov chains, samples sufficiently spaced out over time are nearly independent, and one can potentially use coupling arguments to translate i.i.d. results to corresponding Markov results. Is there any particular challenge that shows up while handling Markov noise in the average reward case that does not show up for the discounted setting? I would presume not.

---

> ### Author Rebuttal · Authors · 2025-07-31
>
> We thank the reviewer for the constructive feedback.
>
> Weakness
>
> (i) As the reviewer pointed out, [1,2] have studied the anchoring mechanism in tabular setting. However, it was not known whether the anchoring mechanism provides benefit in the offline RL setting, where only given dataset is available and distribution mismatch between the data distribution and the transition matrices could cause problem. Moreover, it was not clear whether the anchoring mechanism could be incorporated with general function that approximate the output of the Bellman operator in a supervised learning manner. In this paper, we address these questions and, for the first time, establish finite time bounds for the offline average reward setting for both IID and single-trajectory datasets.
>
> (ii) We first note that in the tabular setup, the iterates of Value Iteration (VI) asymptotically behave as $V^k \sim \Theta(k)$ as $k \rightarrow \infty$ where $V^k$ is $k$-th iterate [3]. Since in our framework, the general function $f_k$ attempts to approximate $V^k$ with offline data, considering the tabular setup as a baseline for the offline setup, we believe that the increasing function range assumption $\|f_k\| \sim \Theta(k)$ is acceptable.
>
>  To address this divergence of the iterates of VI, [4] proposed relative VI which normalizes the iterates by subtracting a uniform constant vector at each iteration. Motivated by this, we propose Relative Anchored Fitted Q-Iteration with function range assumption, specifically assuming $\|f_k\| \sim \Theta(1)$, and as shown in Theorems 3 and 4, we obtained finer sample complexities $\tilde{O}(\epsilon^{-4})$ and $\tilde{O}(\epsilon^{-8})$ for IID and single-trajectory dataset, respectively.
>
> (iii) As reviewer pointed out, in our Theorems 1 and 2, the policy error is bounded by a given $\epsilon$ and an $O(K)$ term containing the coverage coefficient and inherent Bellman error, where, by condition of Theorem, $K$ is fixed if $\epsilon$ is fixed. We note that if the inherent Bellman error is low or the approximation power of the function class is strong enough, this term can be relatively small. Or, it can be ignored under the Bellman completeness assumption, which assumes that the inherent Bellman error is zero. These assumptions, low inherent Bellman error and Bellman completeness, are commonly used in the RL theory literature [5, 6]. Furthermore, we believe this $O(K)$ term in the theoretical upper bound is natural in average reward analysis (please refer to response (v)).
>
> (iv) The reviewer raises a good point. We first note that in section 5, we presented Relative Anchored Fitted Q-Iteration and further obtained refined sample complexities of $\tilde{O}(\epsilon^{-4})$ and $\tilde{O}(\epsilon^{-8})$ for IID and single-trajectory datasets. Also, to the best of our knowledge, there exist two prior works that studied the sample complexity in this setup. [7] and [8] obtained $\tilde{O}(\epsilon^{-2})$ and $\tilde{O}(\epsilon^{-4})$ sample complexities for IID datasets under ergodic and unichain MDPs, respectively, and our Theorem 3 provides a $\tilde{O}(\epsilon^{-4})$ sample complexity for IID datasets under weakly commuting MDPs. But none of prior works address single-trajectory datasets.
>
> As far as we know, there exist no lower bound results in the offline average reward setting. Hence, it is subtle to argue the tightness and optimality of our sample complexity results. However, reflecting optimality results in the discounted reward setup, we expect that our results may be tight or close to optimal sample complexity with respect to $\epsilon$. To be more precise, we speculate that the optimal sample complexity of the average reward setup would be the square of the optimal sample complexity of the discounted reward setup.
>
> In the discounted reward setup, [9, 10] established a $\tilde{O}(\epsilon^{-2})$ lower bound on sample complexity for IID datasets, and several works achieve optimal sample complexity. For the single-trajectory dataset, to the best of our knowledge, there exist no lower bound results, but [11] establishes $\tilde{O}(\epsilon^{-4})$ sample complexity. Considering the technical difficulty in the average reward setup (please refer to response (v)), we carefully expect that our sample complexity results in the average reward setup, $\tilde{O}(\epsilon^{-4})$ and $\tilde{O}(\epsilon^{-8})$ for IID and single-trajectory datasets, which are exactly the squares of the sample complexities in the discounted reward setup, $\tilde{O}(\epsilon^{-2})$ and $\tilde{O}(\epsilon^{-4})$, would be tight or close to optimal. But, as we said, a lower bound in the offline average reward setting remains unknown, and studying optimality in this setup would be an interesting research direction.
>
> (v) To the best of our knowledge, in the discounted reward offline RL setting, there exists an $\epsilon^{-2}$ gap between IID and mixing single-trajectory datasets. Roughly speaking, [5] provides a sample complexity $\tilde{O}(\epsilon^{-2})$ for an IID dataset, whereas [11] proves a sample complexity $\tilde{O}(\epsilon^{-4})$ for a mixing single-trajectory dataset. This discrepancy arises from the different concentration inequalities used in the analyses. In particular, [5] applies Bernstein’s inequality, while [11] uses Pollard’s tail inequality generalized to mixing single-trajectory data, and Pollard’s tail inequality yields looser bounds than Bernstein’s inequality. This induces an $\epsilon^{-2}$ gap between IID and mixing single-trajectory datasets.
>
> Turning to the average reward setup, in the proofs of our Lemmas 5 and 11, we also apply Bernstein’s inequality and the generalized Pollard inequality. Thus, similarly, we have an $\epsilon^{-2}$ gap. But, moreover, an additional gap occurred for the following reason: in the discounted reward setup, the sum of sample errors is bounded by $\sum_{i=1}^{K}\gamma^{i}\epsilon\le \frac{\epsilon}{1-\gamma},$ where $\epsilon$ is the per-iteration sample error. However, in the average reward setup, where the discount factor is considered equal to $1$, we get $O(K)$ bound on the sum of sample errors proportional to the number of iterations by $\sum_{i=1}^{K}\epsilon \le K\epsilon.$ This caused an additional gap in the average-reward setup, and consequently, our results exhibit an $\epsilon^{-4}$ gap, which is the square of the $\epsilon^{-2}$ gap in the discounted reward setup.
>
> Lastly, [12] indeed establishes a sample complexity $\tilde{O}(\epsilon^{-2})$ for both IID and single‑trajectory datasets. However, that paper studies a different setting from ours in two respects. First, Algorithm 1 performs online updates, sampling state and action in real time, so its theoretical analysis does not require the coverage condition that offline RL usually needs to address the distribution shift [5,11]. Second, the paper considers the policy evaluation problem of estimating $V^{\pi}$ for a given policy $\pi$, whereas we study the policy control problem of finding the optimal policy $\pi^{\star}$.  We think that the different results follow from different setups.
>
> (vi) First, we clarify that our work does not assume a mixing MDP for either the IID or single‑trajectory setting.  Actually, we assume only that the single-trajectory dataset, generated by a specific behavior policy, is mixing. As noted in the introduction, a mixing MDP requires the mixing condition to hold for every policy and initial distribution, whereas our mixing assumption requires it to hold only for the behavior policy and the distribution of the dataset.  Thus, our mixing assumption on single-trajectory datasets is much weaker than the mixing assumption on the MDP, and therefore, sample complexity results in the discounted reward setting under mixing Markov chains do not imply our results in either the IID or single-trajectory setting.
>
> We kindly ask that the reviewer consider raising the score if our answer addresses their concerns and questions.
>
> References
>
> [1] J. Lee and E. Ryu. Accelerating value iteration with anchoring. Neural Information Processing Systems, 2023.
>
> [2] J. Lee and E. Ryu. Optimal non-asymptotic rates of value iteration for average-reward MDPs International Conference on Learning Representations, 2025.
>
> [3]  M. L. Puterman. Markov Decision Processes: Discrete Stochastic Dynamic Programming. John Wiley and Sons, 2014
>
> [4] D. J. White. Dynamic programming, Markov chains, and the method of successive approximations. J. Math. Anal. Appl, 6(3):373–376, 1963.
>
> [5] J. Chen and N. Jiang. Information-theoretic considerations in batch reinforcement learning. International Conference on Machine Learning, 2019.
>
> [6] A. Zanette, A. Lazaric, M. Kochenderfer,  E. Brunskill,  Learning near optimal policies with low inherent bellman error. International Conference on Machine Learning 2020.
>
> [7] A. Ozdaglar, S. Pattathil, J. Zhang, and K. Zhang. Offline reinforcement learning via linear programming with error-bound induced constraints. arXiv preprint arXiv:2212.13861, 2024.454
>
> [8] G. Gabbianelli, G. Neu, M. Papini, and N. M. Okolo. Offline primal-dual reinforcement learning for linear MDPs. International Conference on Artificial Intelligence and Statistics, 2024
>
> [9] G. Li, et al. Settling the sample complexity of model-based offline reinforcement learning. The Annals of Statistics 52.1 (2024): 233-260.
>
> [10] P. Rashidinejad, et al. Bridging offline reinforcement learning and imitation learning: A tale of pessimism. Advances in Neural Information Processing Systems 34 (2021): 11702-11716.
>
> [11] A. Antos, C. Szepesvári, and R. Munos. Learning near-optimal policies with bellman-residual minimization based fitted policy iteration and a single sample path. Machine Learning, 2008.
>
> [12] J. Bhandari, D. Russo,  R. Singal. A finite time analysis of temporal difference learning with linear function approximation. Conference on learning theory, 2018.

---

> > ### Comment · Reviewer_vmhP · 2025-08-03
> > **Thank you for rebuttal.**
> >
> > Dear Authors,
> >
> > Thank you for your rebuttal that provides clear answers to all my questions. I am satisfied with the responses, and will raise my score to '4'.

---

### Official Review · Reviewer_bu9m · 2025-07-02

**Clarity:** 4
**Significance:** 4
**Originality:** 3
**Rating:** 4
**Confidence:** 3

**Summary:**

The paper considers the task of offline RL in average-reward MDP with general function approximation. In this setting, the authors propose two “anchored” variants of FQI for learning under a much weaker weakly communicating MDP assumption. Their first algorithm, Anchored FQI utilises an anchoring mechanism to stabilise the original FQI update, by mixing $\hat{f}\_{k+1}=T^* f\_{k}$ the $Q^*$ estimate at each step $k$ with the initial guess $f_0$ with a time-dependent mixing parameter. For this method, they prove $O(\varepsilon^{-6})$, and $O(\varepsilon^{-10})$ sample complexity guarantees when the offline dataset is generated i.i.d, and from a single trajectory respectively. Lastly, they introduce a  Relative Anchored FQI algorithm, which improves the sample complexity results to $O(\varepsilon^{-4})$ for the i.i.d setting and $O(\varepsilon^{-8})$ for the single-trajectory setting, by normalizing $\hat{f}_{k+1}$ in the Anchored FQI scheme. In contrast to related works, their results all hold under a weaker Bellman completeness condition and strong uniform coverage condition.

**Questions:**

1. Can the authors comment on why the uniform coverage condition is necessary for the analysis?

**Ethical Concerns:**

["NO or VERY MINOR ethics concerns only"]

**Final Justification:**

I maintain my positive score of the paper as the strengths outweigh the highlighted weaknesses. More so, this work sheds light on the possibility of developing sample-efficient methods for offline learning in weakly communicating average reward MDPs under general function approximation.

**Quality:**

3

**Strengths And Weaknesses:**

The paper considers a relevant problem of sample-efficient offline learning in weakly communicating average reward MDPs under general function approximation. To achieve this, they uniquely combine tricks from anchoring – an optimization-inspired mechanism with recent success in RL, and relative VI – a provably convergent VI variant for average-reward MDPs. Yet, the algorithms and sample complexity results are dependent on really strong assumptions. First, both algorithms, like FQI, are oracle-efficient as they will require an optimization oracle to compute the sequence of iterates $\\{\hat{f}\_{k}\\}_{k=0}^{K-1}$. While the focus of the paper is not on computational complexity, such complexity results are equally relevant in MDPs with finitely many states – which is the setting considered in the paper. Second, the sample complexity results depend on strong assumptions such as the uniform coverage – which is less accepted in recent literature even under the general function approximation framework due to needing the behaviour policy to properly explore the entire state-action space.

To conclude, despite the weaknesses listed above, I would propose acceptance of this paper on the grounds that it indeed is the first (to the best of my knowledge) to establish sample complexity results for average-reward MDPs under general function approximation and with the milder weakly communicating MDP assumption.

---

> ### Author Rebuttal · Authors · 2025-07-31
>
> We thank the reviewer for insightful feedback.
>
> Questions
>
> 1. Our analysis of anchored fitted Q‑Iteration uses the full coverage condition to bound the products of transition matrices induced by policies. As shown in the proof of Proposition 1, the policy error of approximate anchored Q‑Iteration is bounded by a sum of such products with products of coefficients due to the the anchoring mechanism. To control these terms, we exploit the full coverage coefficient and thereby obtain finite sample complexity. We note that this is how standard fitted Q‑Iteration analyses in the discounted reward setting use the full coverage condition [1,2], and lastly, we briefly note that our anchoring mechanism might provide finite sample complexity guarantees under partial coverage conditions by incorporating a pessimistic approach [3-5].
>
> References
>
> [1] J. Chen and N. Jiang. Information-theoretic considerations in batch reinforcement learning. International Conference on Machine Learning, 2019.
>
> [2] R. Munos and C. Szepesvári. Finite-time bounds for fitted value iteration. Journal of Machine Learning Research, 2008
>
> [3] Yao Liu, Adith Swaminathan, Alekh Agarwal, and Emma Brunskill. Provably good batch off-policy reinforcement learning without great exploration. Neural Information Processing Systems, 2020
>
> [4] T. Xie, C.-A. Cheng, N. Jiang, P. Mineiro, and A. Agarwal. Bellman-consistent pessimism for offline reinforcement learning. Neural Information Processing Systems, 2021
>
> [5]  Y. Jin, Z. Yang, and Z. Wang. Is pessimism provably efficient for offline RL? International Conference on Machine Learning, 2021

---

> > ### Comment · Reviewer_bu9m · 2025-08-03
> >
> > I thank the authors for their response during the rebuttal period and will maintain my positive score as the strengths of the paper outweigh its weaknesses.

---

### Official Review · Reviewer_Uxsp · 2025-07-03

**Clarity:** 3
**Significance:** 3
**Originality:** 3
**Rating:** 5
**Confidence:** 4

**Summary:**

This paper introduces Anchored Fitted Q-Iteration, a novel algorithm for offline average-reward RL with function approximation. The key innovation is the integration of an anchor mechanism, interpreted as weight decay, into fitted Q-iteration, which enables finite-time convergence analysis under weakly communicating MDPs, a significantly milder assumption than the ergodicity/linearity requirements in prior work.  The paper introduces the first finite-time sample complexity bounds for average-reward offline RL with general function approximation, and improves bounds via Relative Anchored Fitted Q-Iteration, leveraging relative normalization.

**Questions:**

- Briefly address whether the analysis could extend to partial coverage (e.g., via [91]) under additional assumptions.
- Can the author briefly discuss the main difficulties of getting an optimal order bound under this setting?

**Ethical Concerns:**

["NO or VERY MINOR ethics concerns only"]

**Final Justification:**

Based on the discussions, I am inclined to recommend acceptance of the paper.

**Limitations:**

yes

**Quality:**

3

**Strengths And Weaknesses:**

**Strengths**

- The paper resolves an open problem by providing the first finite-time guarantees for average-reward offline RL under weakly communicating MDPs. The anchor mechanism is elegantly motivated by Halpern iteration and prior tabular RL work.

- The analysis is thorough, with clear assumptions (e.g., Bellman optimality equation, star-shaped function spaces) and detailed proofs (provided in the appendix).

- The relative normalization mechanism reduces sample complexity by orders of magnitude.

**Weaknesses**
 - Assumptions 5–6 require "full" uniform coverage, which is more restrictive than "partial" coverage in some prior work. While necessary for the analysis, this limits applicability to datasets with insufficient state-action exploration.

- Despite improvements, the bounds are far away from the optimal order in the offline RL setting.

- No empirical validation: While purely theoretical, ablation studies (e.g., anchor vs. no anchor) on synthetic MDPs could strengthen practical insights.

**some comments**
Assumption 1  is not necessary; this is the key for solving average reward MDPs.

---

> ### Author Rebuttal · Authors · 2025-07-31
>
> We thank the reviewer for the positive feedback.
>
> Questions
>
> 1. We expect that our anchoring mechanism can provide finite sample complexity guarantees under partial coverage condition by incorporating a pessimistic approach. A series of works [1–3] has shown that penalizing the estimated value or policy according to its uncertainty relaxes the need for full coverage condition. In particular, [3] introduced Pessimistic Value Iteration, which subtracts an uncertainty term scaled by a confidence parameter from the estimated Q‑value, and achieve finite bounds under partial coverage condition. Referring to those prior works, by combining our anchoring mechanism with this pessimistic approach, we believe that our analysis can be extended to the partial coverage setting.
>
> 2. In the proofs of our Theorems, we observed that the sample error incurred at each iteration grows proportionally with the number of iterations and  $O(K)$ terms appeared in the upper bound of policy error, where $K$ is number of iterations. In contrast, in the discounted‑reward setting, sum of sample error is bounded by $\sum_{i=1}^{K}\gamma^{i}\epsilon \le \frac{\epsilon}{1-\gamma}$ where $\epsilon$ is the per‑iteration sample error (note that the discount factor is considered equal to $1$ in average reward setup). Adding the complexity of Bellman equation, we believe that the sample error in the average reward setup is harder to control. Nevertheless, as we noted in the conclusion, more efficient sample complexity bounds might be obtained by applying variance reduction techniques [4-6] .
>
> Some comments
>
> 1. We briefly note that Assumption 1 is valid for weakly communicating MDPs, and a different assumption is needed for multichain MDPs. For multichain MDPs,  the most general class of MDP, it is known that  the Bellman equations consists of two coupled equations and requires more complicated analysis [7].
>
> References
>
> [1] Yao Liu, Adith Swaminathan, Alekh Agarwal, and Emma Brunskill. Provably good batch off-policy reinforcement learning without great exploration. Neural Information Processing Systems, 2020
>
> [2] T. Xie, C.-A. Cheng, N. Jiang, P. Mineiro, and A. Agarwal. Bellman-consistent pessimism for offline reinforcement learning. Neural Information Processing Systems, 2021
>
> [3]  Y. Jin, Z. Yang, and Z. Wang. Is pessimism provably efficient for offline RL? International Conference on Machine Learning, 2021
>
> [4] M. J. Wainwright. Variance-reduced Q-learning is minimax optimal. arXiv preprin arXiv:1906.04697, 2019.498
>
> [5] J. Lee, M. Bravo, and R. Cominetti. Near-optimal sample complexity for MDPs via anchoring. Interantional Conference on Machine Learning, 2025
>
> [6] A. Sidford, M. Wang, X. Wu, and Y. Ye. Variance reduced value iteration and faster algorithms for solving Markov decision processes. Naval Research Logistics, 70(5):423–442, 2023.
>
> [7] M. L. Puterman. Markov Decision Processes: Discrete Stochastic Dynamic Programming. John Wiley and Sons, 2014

---

> > ### Comment · Reviewer_Uxsp · 2025-08-03
> >
> > I thank the authors for their response. I don't have further questions.

---

### Comment · Area_Chair_XE5m · 2025-08-02

Dear Reviewers,

Please take a look at the authors' response and discuss if there are still more concerns that need to be clarified.
Thanks
AC

---

### Decision · Program_Chairs · 2025-09-17

**Decision:**

Accept (poster)

**Comment:**

This paper develops a new algorithm, Anchored Fitted Q-Iteration, for offline average-reward RL. The key innovation is the integration of an anchor mechanism, interpreted as weight decay, into fitted Q-iteration, which enables finite-time convergence analysis under weakly communicating MDPs, a significantly milder assumption than the ergodicity/linearity requirements in prior work. The paper introduces the first finite-time sample complexity bounds for average-reward offline RL with general function approximation, and improves bounds via Relative Anchored Fitted Q-Iteration, leveraging relative normalization. Some weaknesses are also noted during the review: strong assumptions of full coverage in offline setting, lack of experiments, loose sample complexity bounds.